# Numerical Investigation of Pre-Stressed Reinforced Concrete Railway Sleeper for High-Speed Application

**Zoltán Major** [1], **Sarah Khaleel Ibrahim** [1], **Majid Movahedi Rad** [1], **Attila Németh** [1], **Dániel Harrach** [1], **Géza Herczeg** [1], **Szabolcs Szalai** [1], **Szabolcs Kocsis Szürke** [1], **Dóra Harangozó** [1], **Mykola Sysyn** [2], **Dmytro Kurhan** [3], **Gusztáv Baranyai** [1], **László Gáspár** [4] and **Szabolcs Fischer** [1,*]

[1] Central Campus Győr, Széchenyi István University, H-9026 Győr, Hungary
[2] Department of Planning and Design of Railway Infrastructure, Technical University Dresden, D-01069 Dresden, Germany
[3] Department of Transport Infrastructure, Ukrainian State University of Science and Technologies, UA-49005 Dnipro, Ukraine
[4] KTI Institute for Transport Sciences Non-profit Ltd., H-1119 Budapest, Hungary
* Correspondence: fischersz@sze.hu; Tel.: +36-(96)-613-544

**Abstract:** The current paper deals with the numerical investigation of a unique designed pre-stressed reinforced concrete railway sleeper for the design speed of 300 km/h, as well as an axle load of 180 kN. The authors applied different methodologies in their research: traditional hand-made calculations and two types of finite element software. The latter were AxisVM and ABAQUS, respectively. During the calculations, the prestressing loss was not considered. The results from the three methods were compared with each other. The hand-made calculations and the finite element modeling executed by AxisVM software are adequate for determining the mechanical inner forces of the sleeper; however, ABAQUS is appropriate for consideration of enhanced and sophisticated material models, as well as the stress-state of the elements, i.e., concrete, pre-stressed tendons, etc. The authors certified the applicability of these methodologies for performing the dimensioning and design of reinforced concrete railway sleepers with pre-stressing technology. The research team would like to continue their research in an improved manner, taking into consideration real laboratory tests and validating the results from FE modeling, special material models that allow calculation of crackings and their effects in the concrete, and so that the real pattern of the crackings can be measured by GOM Digital Image Correlation (DIC) technology, etc.

**Keywords:** railway; reinforced concrete; sleeper; prestressing; numerical modeling

## 1. Background

The importance of transport has been an evident since the emergence and development of human societies [1,2]. The most significant changes can be linked to specific dates in world history. Examples are the invention of the wheel, shipping, the industrial revolution, and the steam engine, electricity and internal combustion engines, nuclear power, etc. The transport sciences currently include transport engineering, logistics and transport packaging, civil engineering, electrical engineering, mechanical engineering, vehicle engineering, computer science, and economics, etc. [3–5].

Naturally, when discussing this subject, it is necessary to mention the three main directions within transport, which include and necessarily touch upon the above-mentioned sub-disciplines, forming a common intersection with them. These are land transport, air transport (which includes space transport in this context) and water transport (shipping).

In each of these areas, sustainability, green operation, and the economy are of major importance [6,7].

The present article deals with rail transport within land transport, and within this, with railways (from a civil engineering point of view).

In the case of traditional permanent ballasted railways, their structure consists of the superstructure, i.e., rails, sleepers, rail fasteners (fastening system), the ballast bed with its sub-ballast, as well as the substructure, i.e., granular protection layer (formation layer or protection layer) and the subgrade [8–12]. It has to be mentioned that the sub-ballast is related to the superstructure in European terminology, and it refers to the lower layer of the ballast bed below the sleepers; however, the American terminology follows the method whereby the sub-ballast is a synonym for the formation layer and is categorized as the part of the substructure. The elements of the layer structure compose a (force) support system where and inside it the vertical and horizontal loadings are distributed based on a given distribution law.

The current paper is closely connected to the pre-stressed concrete sleepers. It is the reason why the authors deal only with these structural elements of the ballasted railway tracks. The other parts are only tangentially discussed.

Rails are supported by so-called sleepers (in American terminology, the word tie or cross tie is also applied for most timber or wooden sleepers). Railway sleepers contribute to the distribution of vertical load and lateral movements induced by trains coming from the sleepers directly and via rail fasteners.

The functions and purposes of the sleepers are introduced with the help of the corresponding author's book [13].

The functions of sleepers are:

- to sustain forces from rails and transfer them to the ballast bed with the possible highest uniform manner;
- to provide support and fixing possibilities for the rail foot and fastenings;
- to ensure suitable electrical insulation between rails;
- to guarantee track gauge and rail inclination (mainly 1:20 and 1:40);
- to be resistant to mechanical effects and weathering during their lifetime.

The purposes of sleepers are summarized as below:

- to distribute and transmit forces to the ballast (perpendicular axle loads, longitudinal forces within the rails, centrifugal horizontal forces);
- to establish and maintain the track gauge;
- to hold the rails in height as well as a longitudinal direction;
- to secure the track under construction;
- to dampen rail vibrations;
- to reduce the influence of impact waves and sounds on the environment.

The sleepers, according to their arranging in tracks, can be of the following types:

- twin-block sleeper;
- "longitudinal" sleeper;
- monoblock sleeper;
- slab sleeper;
- frame sleeper;
- other special sleepers, e.g., turnout sleeper, etc.

The sleepers can be classified according to material aspects:

- timber (or wooden) sleepers:
  - hardwood sleepers (e.g., beech, oak, tropical varieties);
  - softwood sleepers (e.g., pinewood);
- steel sleepers;
- concrete sleepers;
  - reinforced concrete sleepers (RC sleepers);
  - pre-stressed reinforced concrete sleepers (PRESRC sleepers);
  - post-stressed reinforced concrete sleepers without bonding technology (POST-SRC sleepers without bonding);

- o　　post-stressed reinforced concrete sleepers with bonding technology (POSTSRC sleepers with bonding);
- plastic and/or composite sleepers [14];
- other special, modern materials.

Pre-stressed reinforced concrete is a methodology that is able to supplement the natural weakness of concrete related to tension. It is applied to produce beams, floors or bridges next to the railway track application, as well. Prestressing tendons (generally of high tensile steel cable or rods) are used to ensure a clamping load which produces a compressive stress that balances the tensile stresses during real loading. Traditional reinforced concrete is based on the use of steel reinforcement bars, rebars, inside poured concrete. Prestressing can be accomplished in three ways: pre-tensioned concrete, and bonded or unbonded post-tensioned concrete.

PRESRC sleepers are applied due to the higher strength and the ensured relatively uncracked state compared to the normal, traditional RC sleepers. The POSTSRC sleepers have not become as popular as much as the PRESRC sleepers because of the relatively complicated production technology.

RC is a composite material in which the concrete possesses low tensile strength and ductility; however, the steel bars (i.e., reinforcement, e.g., like tendons) own higher tensile strength and/or ductility. The reinforcement is usually steel, but other adequate materials can be also applied, e.g., polymer or carbon fibers, etc. The reinforcement in the concrete helps with avoiding the cracks on the tensioned side of the structure, or perhaps with the decreasing of the crack width. The reinforcement can be so called passive or active. Passive means (in this context), for example, reinforcement bars without pre-tensioning before or during the concreting, and they are the so-called normal RC structures. Active reinforcement is the pre- or post-tensioned methodologies where the concrete gets extra, additional pressure via the pre- or post-tensioned reinforcement elements inside it. With active reinforcement, the strength of the RC structure can be increased, while the deformation (mainly deflection) is able to be reduced. The pre- or post-tensioned RC structures are mainly without cracks during their lifetime with service loads.

The international literature is very rich in research related to concrete sleepers. The researchers mainly deal with the following aspects:

- increasing the strength of the RC sleepers with the application of different materials such as concrete mixtures or inclusions, geometry, as well as technologies:
  - o　　steel fibers and graphene oxide [15];
  - o　　laminated carbon fiber reinforced polyurethane (L-CFRPU) [16–20];
  - o　　ultra-high performance concrete (UHPC) [21], as well as ultra-high performance fibre-reinforced concrete (UHP-FRC) and reactive powder concrete (RPC) [22] to be able to avoid pre- or post-tensioning;
  - o　　rubber-concrete [23];
  - o　　macro synthetic fiber-reinforced concrete (MSFRC) [24];
  - o　　metal-matrix composites [25];
  - o　　steel fibers with straight and hooked geometry [26];
  - o　　palm oil fuel ash as supplement material (instead) of cement [27];
  - o　　preplaced aggregate concrete [28];
  - o　　ladder tracks [29];
- increasing the lifetime of the RC sleepers considering fatigue [30,31];
- investigation of RC sleepers in the case of an environment with high humidity [32];
- investigation of lateral stiffness of the ballasted railway tracks [33,34];
- investigation of longitudinal stiffness of the ballasted railway tracks [35];
- the development of new laboratory test methods and arrangements for sleeper bending tests [36];
- the application of waste bi-block concrete sleepers for the improvement and reinforcing of railway earthwork [37];

- the consideration of changing the strength and elastic moduli of concretes during the lifespan of the sleepers [38];
- investigation of transition zones between significantly different vertical stiffnesses [39,40];
- examination of the crack growing in concrete sleepers [17–20,23,24,36,38,41];
- investigation of settlements of ballasted tracks with increased sleeper space [42];
- taking into consideration the noise and/or vibration reduction with different sleepers [27];
- development of RC sleepers for a 40 ton axle load [22];
- examination of environmentally friendly production technology [43];
- investigation of fault production of RC sleepers and the possible reasons for it [44];
- development and analysis of RC sleepers of tramway tracks [31];
- analysis of the debonding effect of twin-block slab tracks [45];
- investigation of the 3D modeling of RC sleepers with the consideration of static, dynamic, and impact loads [30,31,33,35,38,40–42,46,47];
- analysis of the features of the stress-strain state of the dual gauge track with a special design of reinforced concrete sleepers with the simultaneous fastening of four rails [48]. The methods listed below have been applied in the papers:
- numerical modeling (finite element method—FEM and/or discrete element method—DEM) [30,31,33,35,38,40–42,46,47];
- numerical modeling (a model of the stressed-strained state of a railway track based on the dynamic problem of elasticity theory) [48–50];
- numerical modeling (MATLAB, etc.) [39];
- laboratory measurements with [23,36,38] or without Digital Image Correlation (DIC) [15–22,24,26–30,34,35,51];
- analytical modeling [30];
- field tests [29,32,33,39,40,42];
- theoretical and practical calculations, as well as calculations of life-cycle costs [43,44,52];
- literature review [53,54].

The above-mentioned papers focus mainly on the improved technologies and methodologies of the RC and PRESRC sleepers. The authors (of the current paper) found, stated, and assumed that the following aspects will be useful for their research (now and in the future): high-strength concretes and their mixtures, macro fibres, fatigue life, and the discrete element modeling (DE modeling) of the lateral resistance of the railway tracks comprised of different sleeper geometries.

In this paper, the authors designed a prestressed reinforced concrete rail cross beam for a design speed of 300 km/h and a static design axle load of 180 kN using engineering literature design procedures. The inner forces of the sleeper were determined in a simplified finite element model (namely the AxisVM X5 software), upon which a sophisticated finite element modeling was also performed (namely with the ABAQUS 2018 software) after specifying the dimensions and material types of the sleeper. No such complex dimensioning and modeling have been published in the international literature. The Methods are presented in Section 2 of the paper, the Results are presented in Section 3, and the Discussion is conducted in Section 4. Section 5 contains the Conclusion. A detailed list of abbreviations and a list of the nomenclature have been placed at the end of the paper.

## 2. Methods

The literature on the practical design of pre-stressed reinforced concrete railway sleepers was last published in Hungary in 1965. The book of the Műszaki Könyvkiadó (i.e., Technical Book Publisher) entitled *Concrete Railway Sleepers* [55] summarized the knowledge available at the technological and regulatory level of the given age. The development of technology, the change in the regulatory background, and the construction of elevated speed railway tracks in Hungary have continuously inspired the development of concrete sleepers. In 1999, János Beluzsár published his article *LW 60, concrete sleepers for high-speed railway tracks* [56]. The author described the steps of sizing the concrete sleepers in this paper, but did not discuss the topic in complete detail; nevertheless, it is

an unavoidable part of Hungarian literature. In 2015, two articles were published in the *Innorail* magazine, whose results can be used effectively to process the topic. R. Fischer's article on pre-stressed concrete sleepers on the ÖBB (Austrian State Railways) network [57] describes the production specifications of concrete sleepers and the factors that determine the sleepers' service life. Kormos et al. [58] drew attention to the up-to-date results of finite element modeling and made several remarks that can be transformed into everyday practice. In this paper, the authors will present a method for determining the mechanical inner forces according to the current regulations and specifications filling in the gaps in the Hungarian literature on the topic. Using some aspects of the calculation method according to the UIC 713 recommendation [59] of the International Union of Railways (UIC), the calculations were executed by AxisVM software.

The authors determined the stresses of a prototype concrete sleeper suitable for a speed of $V = 300$ km/h and an axle load of $Q = 180$ kN. The determined dimensions are fictitious; their role was founded only in the calculation. The geometric design of the investigated cross base is shown in Figure 1.

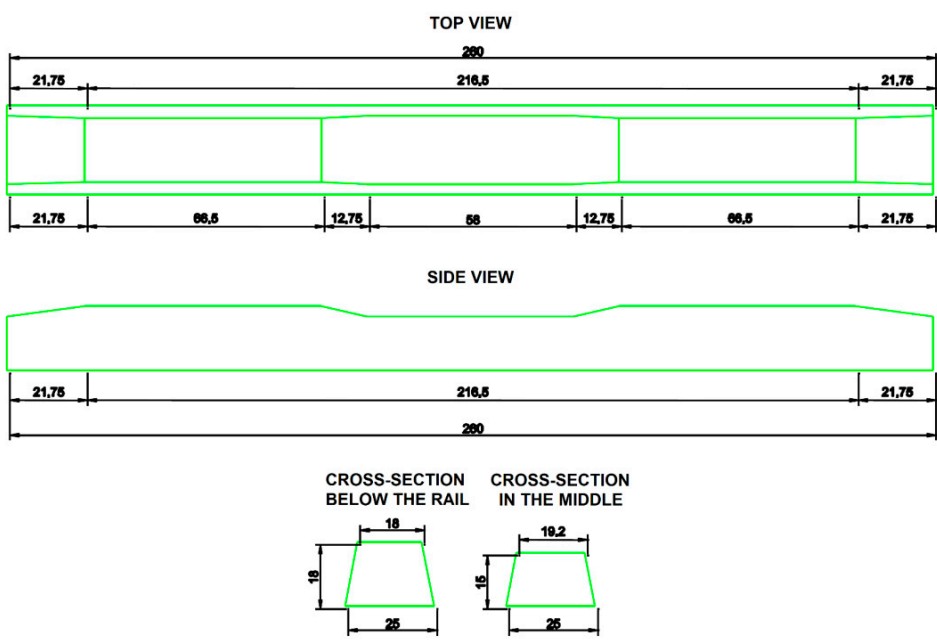

**Figure 1.** Specific dimensions of the prototype railway sleeper (the dimensions are given in centimeter units).

The applied method in the current paper is the finite element method (FE method or FEM). The authors used two different types of FE software during their research: the first is AxisVM, and the other is ABAQUS. AxisVM is a normal, "basic" software that is mainly applied for the solution of max. 2.5D problems. It means that the used model is not a real 3D model; however, it is more than 2D with 3D visualization. Therefore, the results are adequate for standard (civil) engineering applications. Hence, ABAQUS is a real 3D FE software with special material models.

During the research, AxisVM was used for determining mechanical inner forces, i.e., there was a so-called frame structure (model) to obtain the forces transmitted from the rail feet; it was executed with Winkler-type springs as support "structures". Based on the results of the frame structure model, an isolated sleeper model was prepared and generated. The support of the isolated sleeper was line support (or in other words: lengthwise support) that was appropriate for calculating the mechanical inner forces of the structure (i.e., the designed railway sleeper).

ABAQUS software considers and uses enhanced material models; it takes into consideration real 3D models where several structural elements and components (i.e., concrete,

pre-stressed tendons, etc.) are able to be modeled, and the related stress states can be determined and computed.

It has to be mentioned that the change in tension over time is not considered in detail in the article and is not taken into account. The reason is that the quantification of tension losses requires the consideration of a large number of parameters and the knowledge of the exact manufacturing technology. As these were not available, since a fictitious product was modeled, it was more appropriate to modify the final results obtained without taking into account the stress losses. For this purpose, the recommendation of the literature [60] was taken into account, according to which for bridge beams manufactured under Hungarian conditions, the tension losses can be up to 25% of the initial tension stress. Since we do not know the exact parameters here either, the authors approximated this value by 40% in our study. Thus, only 60% of the results obtained were taken into account. This solution gives almost exact results for the cracking moment with a 40% loss of tension force, and for the breaking loads it is strongly biased in favor of safety, the reason being that the effect of tension force on breaking loads is not significant within certain limits, as pointed out in [61] in the section on the breaking of tensioned supports.

The calculation presented here was not intended to quantify all effects, including speeds above 300 km/h, as this is not covered by the UIC recommendation [59]. Of course, the principles presented can be applied in this case as well, but it is up to the designer to choose the appropriate parameters. The finite element part of the design process is of course generally applicable. In summary, with a careful choice of parameters, the method is suitable for the design of cross beams up to 350 km/h.

## 3. Results

### 3.1. Determination of Mechanical Inner Forces of the Sleeper Using AxisVM finite Element Software

The finite element modeling was performed in two steps. In the first step, a simplified track model was applied to determine the value of the pressure acting onto the sleepers (from the rail feet). At this stage of the modeling, the issue of the change in the embedding factor and the examination of possible irregular supports (i.e., support in the middle zone of the sleepers and the evolving negative bending moment in the sleepers) should also be considered. This is because they play a crucial role in the load distribution between sleepers. In the second step, the sleepers' bending moment and shear force graphs were defined on the sleeper's separate "rod model" using the rail pressures in each case. The finite element calculations were then executed using AxisVM X5 software.

The track model consists of two-rod elements with a cross-sectional design. 60E1 rail profile was chosen and applied based on the railway rail catalog of VoestAlpine. As a simplification, the cross-sectional sizes of the sleepers were considered with the dimension characteristics of the cross-section under the rail feet (i.e., rail support on the sleeper). The individual sections and their cross-sectional characteristics are shown in Figures 2 and 3.

The distance between the track rail axes is $t_R$ = 1500 mm in the model. The length of the sleepers is $L_S$ = 2600 mm.

The rails are rigidly connected to the sleepers, considering the offset of the centers of gravity. Eccentricity was achieved by inserting rigid bodies. (In the AxisVM X6 version, manually installing rigid bodies would not have been necessary; during modeling, the bar elements can be connected eccentrically). The material characteristics are illustrated in Table 1. It has to be mentioned that the S235 steel category was only applied for AxisVM X5 finite element modeling, where the inner forces (bending moment/torque, shear forces, and support forces) were calculated. In this case, the steel quality doesn't influence the results. However, in the case of sophisticated modeling, i.e., in ABAQUS, the steel tendons were modeled by high-strength steels (see Section 3.2.3).

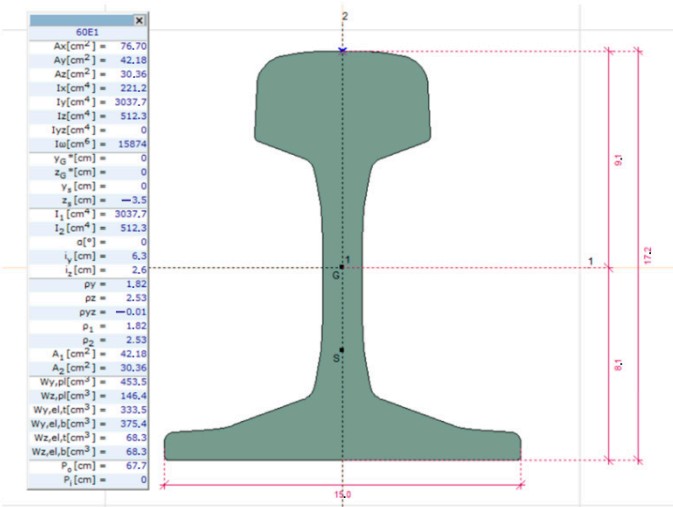

**Figure 2.** Cross-section and the relevant geometrical characteristics of the applied railway rails (for further information and the explanation of the parameters, see the Nomenclature in the end of this paper).

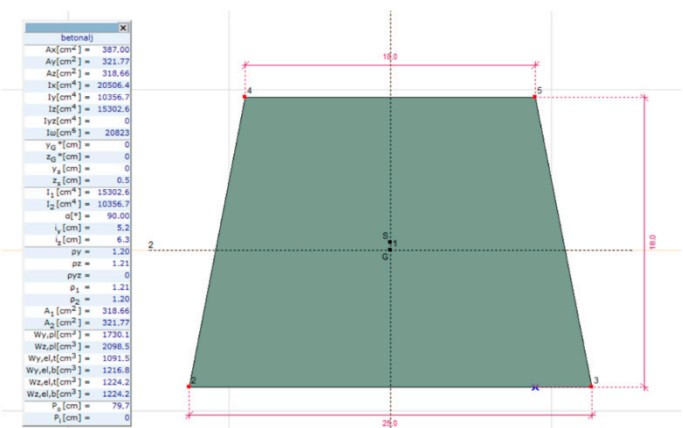

**Figure 3.** Cross-section and the relevant geometrical characteristics of the applied railway sleepers (for further information and the explanation of the parameters, see the Nomenclature in the end of this paper).

**Table 1.** Applied material characteristics.

| Material | Type | National Standard | Material Standard | Model | Young Modulus $E_x$ [kN/cm²] = $E_y$ [kN/cm²] | Poisson Ratio $\nu$ [−] | Linear Thermal Expansion Ratio $\alpha_T$ [1/°C] | Density $\rho$ [kg/m³] |
|---|---|---|---|---|---|---|---|---|
| C50/60 | concrete | Eurocode-H | EN 206 | linear | 3730 | 0.20 | $1 \times 10^{-5}$ | 2500 |
| S 235 | steel | Eurocode-H | 10025-2 | linear | 21,000 | 0.30 | $1.2 \times 10^{-5}$ | 7850 |

In the model, the length of the track is 30 m, and sleeper space is 60 cm. The axonometric view of the track is shown in Figure 4.

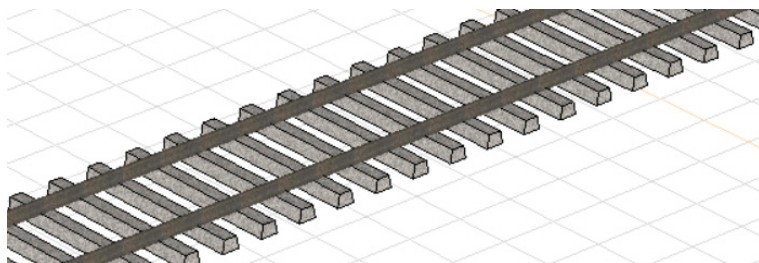

**Figure 4.** Axonometric view of the track considered.

The sleepers are supported vertically by node supports under the rails in the model. In determining the spring constants of the node supports, the effect of the magnitude of the bearing factor and the change in bearing geometry were considered. Based on the authors' experience (in the case of a modern, well-maintained track), a value of $C = 0.20 \text{ N/mm}^3$ can be applied on the interface between the sleepers' bottom and ballast. This value may decrease due to the track's deterioration and the effects of traffic. In addition to the application of under sleeper pads, which is becoming more and more widespread in Hungary, the value of the resulting embedding factor can be as high as $C = 0.05 \text{ N/mm}^3$. During the design, it is expedient to indicate the possibility of the design with the under sleeper pads (USPs), so the embedding factor with the following values was characterized; see Equation (1).

$$C = 0.05 \ldots 0.10 \ldots 0.15 \ldots 0.20 \left[ \frac{\text{N}}{\text{mm}^3} \right] \tag{1}$$

where $C$ is the bedding modulus below the sleeper's bottom.

**Remark 1.** *in the case of softer under sleeper pads, the calculation should be performed for a lower embedding factor, if necessary.*

The surface in contact with the ballast bed (interface) was considered for a freshly tamped track with respect to a rail: 1010 mm × 250 mm. While for the irregular supported sleepers it was 1300 mm × 250 mm. The interpretation of the supporting lengths is shown in Figure 5.

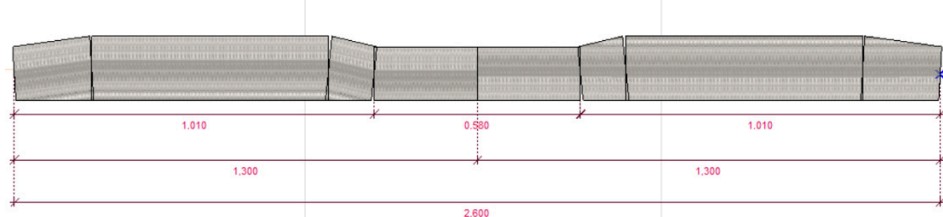

**Figure 5.** Length of support of sleepers (unique model).

The spring constant values applied in the model are summarized in Table 2, which is calculated using Equation (2).

$$K_z = C \cdot A \tag{2}$$

where

- $K_z$: vertical spring constant for one rail [kN/m];
- $A$: supporting contact area regarding one rail [mm$^2$].

**Table 2.** Spring constant values used in the model (per rail).

| $C$ [N/mm³] | $K_z$ [kN/m] Tamped Sleeper Where $A$ = 252,500 [mm²] (250 × 1010) | $K_z$ [kN/m] Irregular Supported Sleeper Where $A$ = 325,000 [mm²] (250 × 1300) |
|---|---|---|
| 0.05 | 12,625 | 16,250 |
| 0.10 | 25,250 | 32,500 |
| 0.15 | 37,875 | 48,750 |
| 0.20 | 50,500 | 65,000 |

In the applied track model, the difference of the vertical wheel loads due to accidental eccentricity was considered. However, the accidental eccentricity of vertical loads can also result from the unequal loading of wagons and the sinusoidal running of wagons. For this reason, the vertical loads must not be evenly distributed between the two rails. In the authors' calculation they followed the principles according to the standard EN 1991-2 [62], according to which the extreme value of the ratio of wheel loads is (see Equation (3)):

$$\frac{Q_{v1}}{Q_{v2}} = \frac{4/9}{5/9} = \frac{1}{1.25} \tag{3}$$

where

- $Q_{v1}$: one of the vertical wheel forces in a wheelset [kN];
- $Q_{v2}$: the other vertical wheel force in the same wheelset [kN];
- $Q_{v1} + Q_{v2}$: vertical axle force (load) in the mentioned wheelset, the sum of them is $Q_v$ [kN].

The accidental (random) eccentricity is presented in Figure 6.

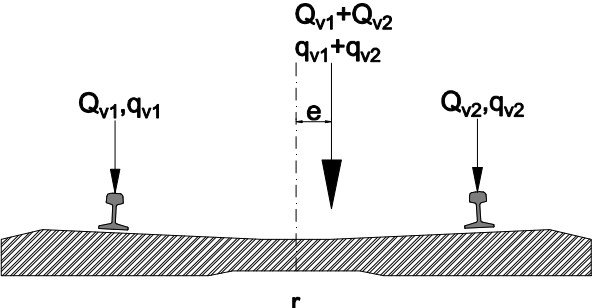

**Figure 6.** Interpretation of the eccentricity of vertical loads in the case of the LM71 load model according to the EN 1991-2 standard [62].

In Figure 6 the meanings of the parameters are the following:

- $e$: the random eccentricity in [m] or in [mm] units;
- $r$: distance between rail centers; its nominal value is 1.5 m if there is a standard track gauge (1.435 m);
- $q_{v1}$ and $q_{v2}$: vertical force acting on each rail [kN].

The value of the static axle load is 180 kN. Based on this value, the static wheel force distribution was determined corresponding to the transverse random eccentricity of the vertical load, based on which the following values were obtained (see Equations (4) and (5)):

$$Q_{v1} = \frac{4}{9} \cdot Q_v = \frac{4}{9} \cdot 180 = 80 \ [\text{kN}] \tag{4}$$

$$Q_{v2} = \frac{5}{9} \cdot Q_v = \frac{5}{9} \cdot 180 = 100 \ [\text{kN}] \tag{5}$$

In the track model, a bogie was considered, i.e., two axle loads spaced 1800 mm apart, as shown in Figure 7. In the model, only an eccentric load was applied.

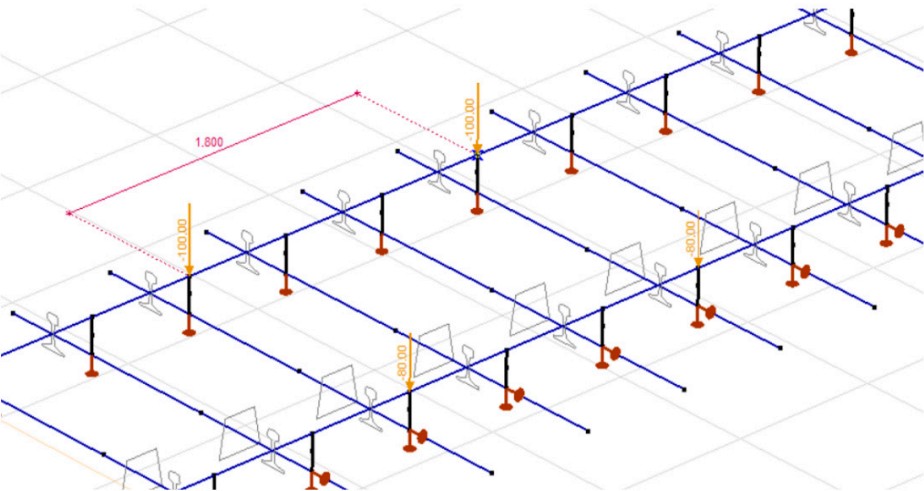

**Figure 7.** The loaded track model.

After performing the calculation, the obtained result is shown in Figure 8 for a spring constant of 12,625 kN/m. The static setpoint of the maximum vertical force acting on the sleepers is 34.03 kN, as shown in Figure 8. The results obtained for the additional cases are summarized in Table 3.

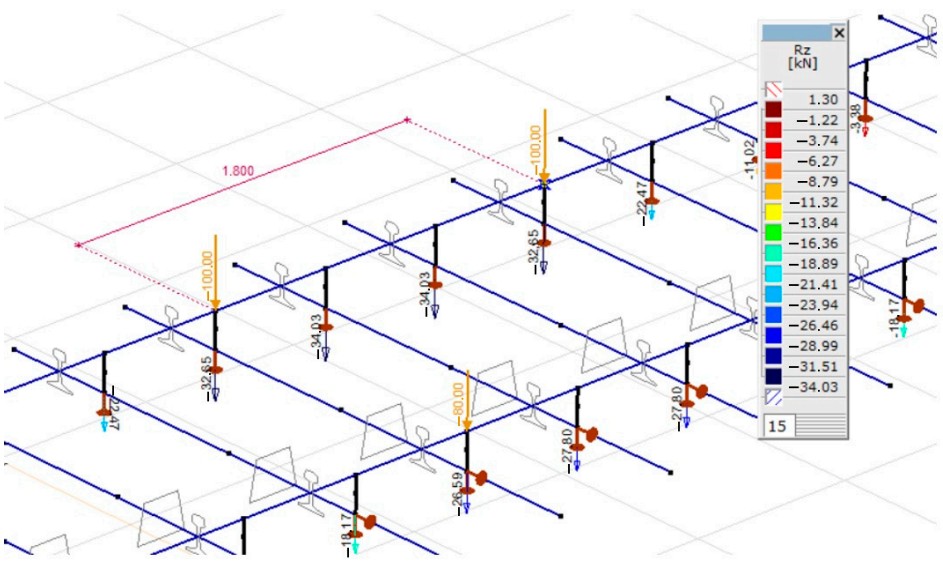

**Figure 8.** The distribution of supporting forces on the track model.

**Table 3.** Maximal vertical forces acting on the sleepers.

| $C$ [N/mm³] | $F_z$ [kN] Tamped Sleeper | $F_z$ [kN] Irregular Supported Sleeper |
|---|---|---|
| 0.05 | 34.03 | 34.33 |
| 0.10 | 35.69 | 37.02 |
| 0.15 | 37.91 | 39.53 |
| 0.20 | 39.78 | 41.67 |

Considering the width of the rail foot and the load distribution in the sleeper up to the neutral axis, the linear distribution load to be applied to the separate sleeper models can be determined. Half of the load distribution width (*f*) can be determined from Equation (6):

$$f = \frac{s+d}{2} = \frac{0.15+0.18}{2} = 0.165 \ [\text{m}] \tag{6}$$

where

- *s*: the width of the rail foot in [m] unit, in the case of the 60E1 rail profile *s* = 0.15 m;
- *d*: the thickness of the sleeper under the rail foot's center in [m] unit, 0.180 m was considered in this paper (this value was determined with the help of safety due to the trapezoid cross-sectional shape).

The double value of *f* can be considered (see Equation (6)) as a load distribution length, i.e., 0.330 m. The linear distribution load can be defined according to Equation (7), and the calculated values are summarized in Table 4.

$$q_Z = \frac{F_z}{2 \cdot f} = \frac{F_z}{2 \cdot 0.165} = \frac{F_z}{0.330} \tag{7}$$

**Table 4.** Maximal linear distributed vertical forces acting onto the sleepers.

| $C$ [N/mm$^3$] | $Q_z$ [Kn] Tamped Sleeper | $q_z$ [kN] Irregular Supported Sleeper |
|---|---|---|
| 0.05 | 103.0 | 104.0 |
| 0.10 | 108.0 | 112.0 |
| 0.15 | 115.0 | 120.0 |
| 0.20 | 121.0 | 126.0 |

In the isolated model, two load cases were examined:

- Case #1: 90% of the specified maximum load ($Q_v{}'$) is applied to both parts under the rail (central load, see Equation (8));
- Case #2: one part below the rail foot receives the maximum load, while the other gets 80% of the load (eccentric load, based on a wheel load distribution of 1.00:1.25 = 0.80:1.00).

$$Q_{v'} = \frac{9}{10} \cdot \frac{5}{9} \cdot Q_v = \frac{5}{10} \cdot Q_v \tag{8}$$

An example of the load modes (patterns) used in the separate model is illustrated in Figures 9 and 10, respectively.

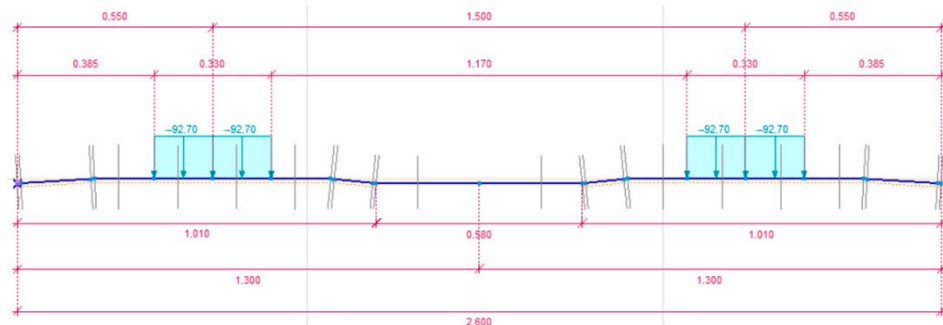

**Figure 9.** Central (Centric) loaded sleeper model.

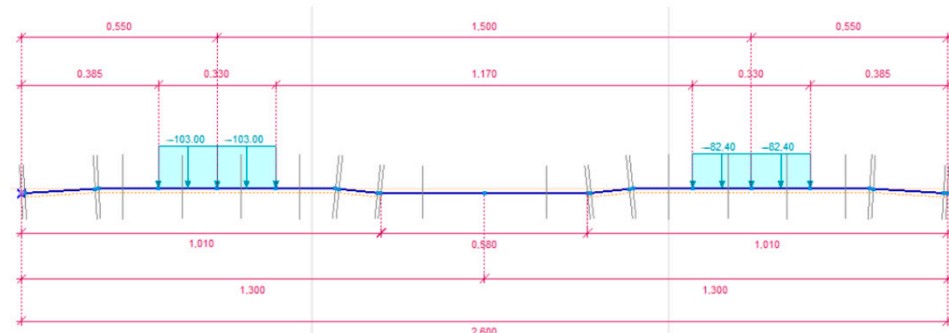

**Figure 10.** Eccentric loaded sleeper model.

The geometric design of the isolated models is shown in Figure 11. The elastic embedded structure is composed of beam elements of variable cross-section, the geometrical dimensions of which are shown in Figure 1. The material quality of the bar elements is C50/60.

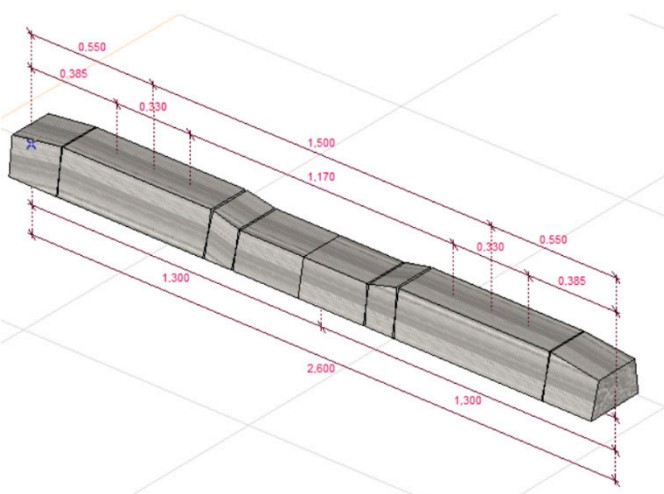

**Figure 11.** The axonometric view of a prototype sleeper.

The linear spring constants applied in the model are summarized in Table 5, which is calculated using Equation (9).

$$k_z = C \cdot b_1 \cdot 1000 \tag{9}$$

where

- $k_z$: longitudinal (or in others word, lengthwise) spring constant in the [kN/m/m] unit;
- $b_1$: width of the sleeper bottom in [mm] unit.

**Table 5.** Longitudinal spring constants related to one rail.

| $C$ [N/mm$^3$] | $k_z$ [kN/m/m] |
| --- | --- |
| 0.05 | 12,500 |
| 0.10 | 25,000 |
| 0.15 | 37,500 |
| 0.20 | 50,000 |

Depending on whether a well-maintained track condition (properly tamped sleepers) or irregular supported sleeper, the support length was considered in the isolated model.

After performing the linear static calculation based on the effect of the static load in the studied and considered cases, the bending moment diagrams of the sleeper became

determinable. A bending moment series is illustrated in Figures 12–15. According to Figures 12–15, it can be seen that the eccentric load case gives less favorable results in all cases, so in Table 6 the bending moments from the eccentric load are summarized.

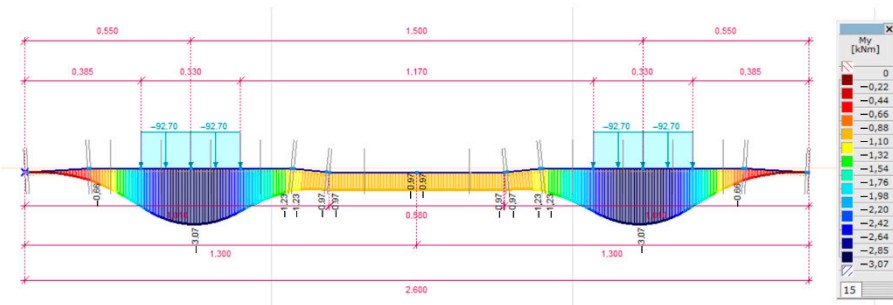

**Figure 12.** Bending moment diagram in the case of $C = 0.05$ N/mm$^3$; a well tamped sleeper and centric loading arrangement.

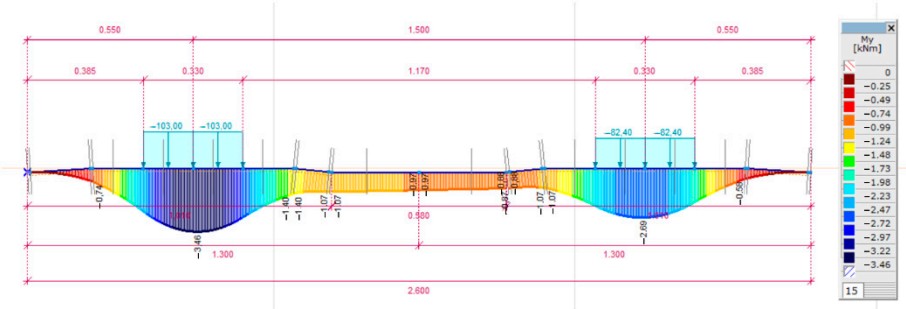

**Figure 13.** Bending moment diagram in the case of $C = 0.05$ N/mm$^3$; a well tamped sleeper and eccentric loading arrangement.

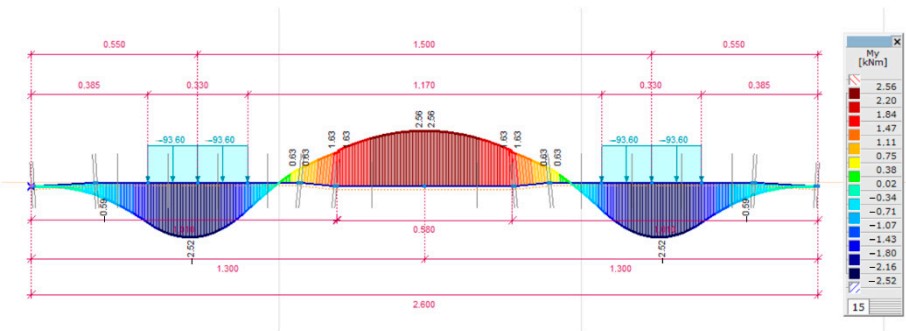

**Figure 14.** Bending moment diagram in the case of $C = 0.05$ N/mm$^3$; an irregular supported sleeper and centric loading arrangement.

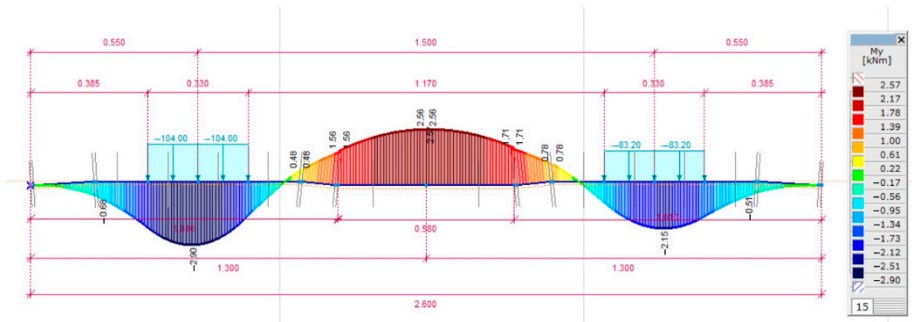

**Figure 15.** Bending moment diagram in the case of $C = 0.05$ N/mm$^3$; an irregular supported sleeper and eccentric loading arrangement.

**Table 6.** Bending moment values of the sleeper in the case of an eccentric loading arrangement.

| $C$ [N/mm³] | Under Rail Foot | | In the Center of the Sleeper | |
|---|---|---|---|---|
| | $M_y$ [Knm] Tamped Sleeper | $M_y$ [kNm] Irregular Supported Sleeper | $M_y$ [Knm] Tamped Sleeper | $M_y$ [kNm] Irregular Supported Sleeper |
| 0.05 | 3.46 | 2.90 | 0.97 | −2.56 |
| 0.10 | 3.44 | 3.21 | 0.74 | −2.44 |
| 0.15 | 3.52 | 3.47 | 0.57 | −2.40 |
| 0.20 | 3.58 | **3.67** | 0.43 | −2.36 |

It can be seen in Table 6 that the maximum bending moment of the sleeper in the present case occurs both under the rail foot and in the center of the sleeper; in both cases the irregular supported sleepers are relevant; under the rail foot for stiffer support and under the center of the rail for softer support. Therefore, the design bending moment values were determined using these two values as a mean.

According to the recommendation of UIC [61], the additional load due to geometrical faults is taken into account, with a value of $\gamma_r = 1.35$. The effect of the support faults along the length of the sleeper is taken into account with a value of $\gamma_i = 1.60$ (as safety factors). The effect of speed (instead of $\gamma_V$) is validated by Eisenmann's multiplication, while the damping effect of the elastic rail pad is neglected in the calculation ($\gamma_p$), because by changing the bedding modulus, its effect is taken into account. The use of a factor $\gamma_d$ that considers the load distribution between the sleepers is unnecessary, as the load distribution is performed automatically by the FE software.

There is an introduced additional unit ($\Gamma$) according to the recommendation of UIC using the multiplication of the considered factors (see Equation (10)).

$$\Gamma = \gamma_r \cdot \gamma_i = 1.35 \cdot 1.60 = 2.16 \tag{10}$$

According to the Eisenmann equation, the value of the speed factor ($\varphi$) can be calculated: it is 1.00 for $V \leq 60$ km/h and based on Equation (11) in the case of $V \leq 200$ km/h.

$$\varphi = 1 + \frac{V - 60}{140} \tag{11}$$

where

- $V$: the allowed speed on the track in [km/h] unit.

Movement with such speeds has no difference in the reaction of the rail between dynamic and static deflection [50]. In Hungarian practice, Equation (11) has been sufficient so far; in the case of speeds $V \geq 200$ km/h, the international literature reports correlations with the modified Eisenmann equation. For example, the following correlations can be found in the book *Ballastless Tracks* [63], as they are shown in Equations (12)–(14). Equation (12) is related to freight trains, while Equation (13) involves passenger trains.

$$\varphi = 1 + \frac{V - 60}{160} \tag{12}$$

$$\varphi = 1 + \frac{V - 60}{380} \tag{13}$$

The calculations are performed considering a 180 kN axle load. This is the reason why Equation (13) is applied, as the results can be seen in Equation (14).

$$\varphi = 1 + \frac{V - 60}{380} = 1 + \frac{300 - 60}{380} = 1 + \frac{240}{380} = 1.63 \tag{14}$$

The value of the dynamic factor depending on the track condition and speed can be calculated using Equation (15); and the results are incorporated in Equation (16).

$$\Phi = 1 + n \cdot t \cdot \varphi \tag{15}$$

where

- $n$: track condition parameter, $n = 0.2$ is considered;
- $t$: Student distribution parameter, 99.7% calculation accuracy is taken into consideration when $t = 3.0$.

$$\Phi = 1 + n \cdot t \cdot \varphi = 1 + 0.2 \cdot 3.0 \cdot 1.63 = 1.98 \cong 2.0 \tag{16}$$

In the general case, the design value of the bending moment can be calculated by Equation (17).

$$M_d = \Phi \cdot \Gamma \cdot M \tag{17}$$

where

- $M_d$: the design value of the bending moment in [kNm] units;
- $M$: the mean value of the bending moment in [kNm] units;
- $\Phi$: dynamic factor based on Equation (16);
- $\Gamma$: based on Equation (10).

The bending moment causing cracking under the rail foot has two extreme values: the positive ($M_{dr+}$) and negative ($M_{dr-}$), respectively (see Equations (18) and (19)).

$$M_{dr+} = 2.00 \cdot 2.16 \cdot 3.67 = +15.85 \text{ kNm} \tag{18}$$

$$M_{dr-} = -\frac{M_{dr+}}{2} = -\frac{15.85}{2} = -7.93 \text{ kNm} \tag{19}$$

The bending moment values causing cracking related to the sleeper's center are shown in Equations (20) and (21).

$$M_{dc-} = 2.00 \cdot 2.16 \cdot (-2.56) = -11.06 \text{ kNm} \tag{20}$$

$$M_{dc+} = -0.7 \cdot M_{dc-} = -0.7 \cdot (-11.06) = +7.74 \text{ kNm} \tag{21}$$

The bending moment values causing breakage ($M_{dr+,breakage}$ and $M_{dc-,breakage}$) are determined by safety factor $k_2 = 2.5$ according to UIC [61]; see Equations (22) and (23).

$$M_{dr+,breakage} = k_2 \cdot M_{dr+} = 2.50 \cdot 15.85 = 39.63 \text{ kNm} \tag{22}$$

$$M_{dc-,breakage} = k_2 \cdot M_{dc-} = 2.50 \cdot (-11.06) = -27.65 \text{ kNm} \tag{23}$$

The calculation was also executed according to the recommendation of UIC, which is not reported and published in this article, but the obtained results are compared with the results from finite element modeling in Table 7.

**Table 7.** Comparison of the results from the calculation based on the recommendation of UIC, as well as FE modeling (AxisVM).

| Parameters | Recommendation of UIC [kNm] | FE Modeling [kNm] | Deviation [%] |
|---|---|---|---|
| $M_{dr+}$ | +14.87 | +15.85 | 6.6 |
| $M_{dr-}$ | −7.44 | −7.93 | 6.6 |
| $M_{dr+,breakage}$ | +37.18 | +39.63 | 6.6 |
| $M_{dc-}$ | −10.71 | −11.06 | 3.3 |
| $M_{dc+}$ | +7.50 | +7.74 | 3.3 |
| $M_{dc-,breakage}$ | −26.78 | −27.65 | 3.3 |

Based on Table 7, it can be seen that the stresses determined by the two design methods correlate well. The maximum deviation is 6.6%.

The calculation of shear forces is not covered by the UIC recommendation [61], but finite element modeling also allows the shear force diagrams to be determined and analyzed in the event of changes in various parameters. In the case of two bedding moduli, the shear force diagram determined from the eccentric load is shown in Figures 16–19. Due to the display of the AxisVM FE software, only the standard half is shown in the figures.

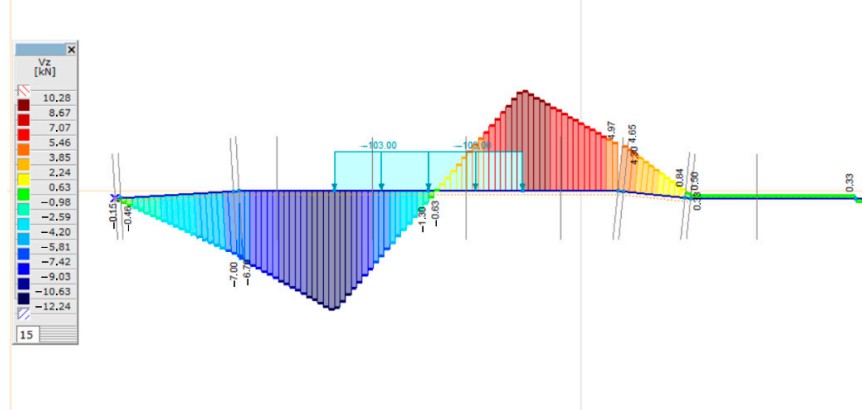

**Figure 16.** Shear force diagram in the case of $C = 0.05$ N/mm$^3$; well tamped sleeper and eccentric loading arrangement.

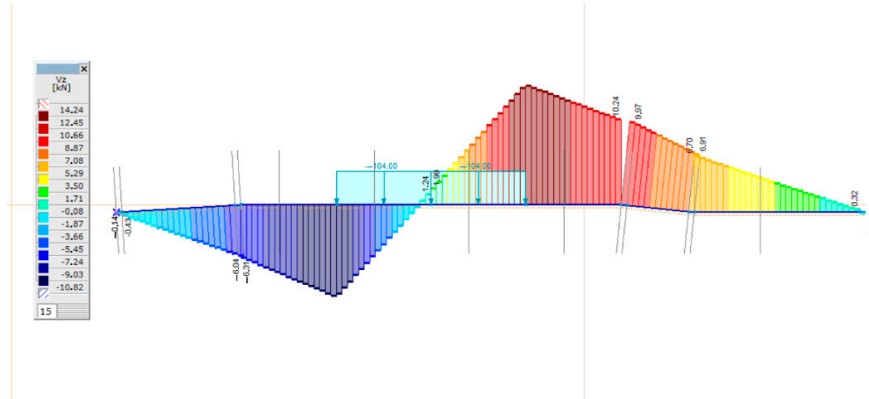

**Figure 17.** Shear force diagram in the case of $C = 0.05$ N/mm$^3$; irregular supported sleeper and eccentric loading arrangement.

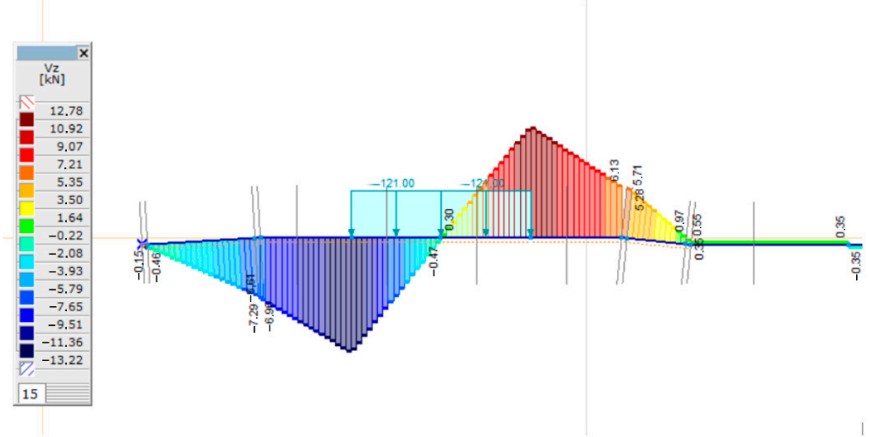

**Figure 18.** Shear force diagram in the case of $C = 0.05$ N/mm$^3$; well tamped sleeper and eccentric loading arrangement.

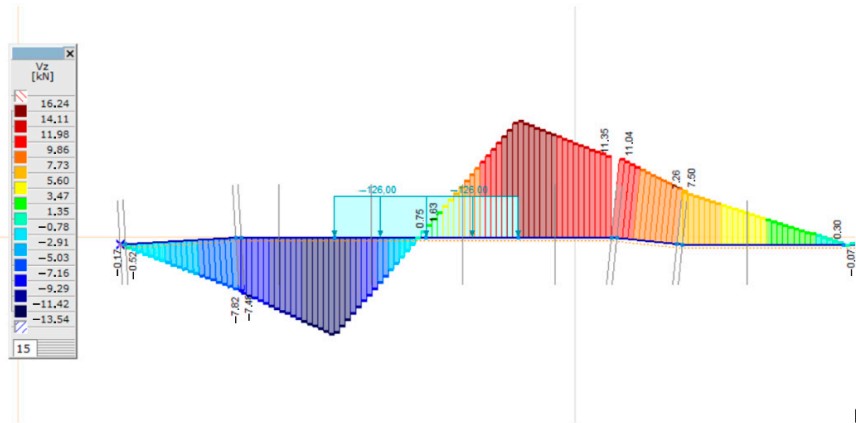

**Figure 19.** Shear force diagram in the case of $C$ = 0.05 N/mm$^3$; irregular supported sleeper and eccentric loading arrangement.

The mean values of the shear forces determined based on the finite element modeling are summarized in Table 8.

**Table 8.** Mean values of shear forces of the sleeper based on finite element modeling.

| $C$ [N/mm$^3$] | Under Rail Foot | |
| --- | --- | --- |
| | $V_z$ [kN] Tamped Sleeper | $V_z$ [kN] Irregular Supported Sleeper |
| 0.05 | 12.24 | 14.24 |
| 0.10 | 12.39 | 14.93 |
| 0.15 | 12.84 | 15.69 |
| 0.20 | 13.22 | **16.24** |

It can be seen in Table 8 that the higher the support stiffness, the higher the shear forces; and higher values are obtained for the irregular supported sleepers than for the well tamped soles. The maximum of the mean shear force is 16.24 kN (see Table 8).

For the design value of shear force, the multipliers are considered similar to the bending moment. The difference is that the UIC recommendation [61] does not use the quantity $\gamma_i$ as a multiplication factor when calculating the load acting below the rail foot. Since the quantity $\gamma_i$ is not applied, the design value of the shear force can be calculated using Equation (24).

$$V_{z.Ed} = \Phi \cdot \gamma_r \cdot V_z = 2.00 \cdot 1.35 \cdot V_z = 2.70 \cdot 16.24 = 43.85 \text{ kN} \tag{24}$$

**Remark 2.** *the UIC recommendation [61] omits the issue of shear calculation, so the used multiplication is a unique concept, not a mature solution based on professional consensus.*

In addition to the determination of shear forces, the great advantage of finite element modeling is that the surface pressure acting on the railway ballast can be determined, for which the UIC recommendation [61] does not provide a calculation method. Figures 20 and 21 illustrate examples of the line support forces determined by the finite element model.

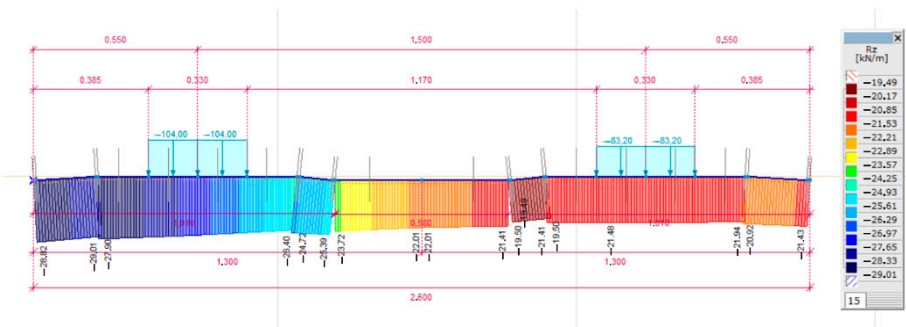

**Figure 20.** Support force diagram in the case of $C$ = 0.05 N/mm$^3$; irregular supported sleeper and eccentric loading arrangement.

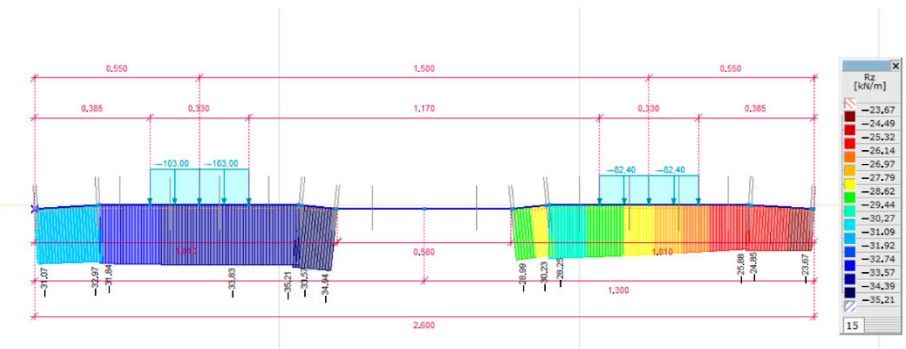

**Figure 21.** Support force diagram in the case of $C$ = 0.05 N/mm$^3$; well tamped sleeper and eccentric loading arrangement.

The maximum values obtained for the mean values of the line support forces ($R_z$) during the modeling are summarized in Table 9.

**Table 9.** Mean values of line support forces of the sleeper based on finite element modeling.

| $C$ [N/mm$^3$] | $R_z$ [kN/m] Tamped Sleeper | $R_z$ [kN/m] Irregular Supported Sleeper |
|---|---|---|
| 0.05 | 35.21 | 29.01 |
| 0.10 | 37.97 | 31.84 |
| 0.15 | 41.14 | 34.45 |
| 0.20 | **43.80** | 36.67 |

It can be seen from Table 9 that the higher the linear support force, the higher the support stiffness, and with the value of the well tamped sleepers, a smaller support surface. The maximum value of the line support force is 43.80 kN/m.

As the support force can be calculated from the load transferred from the rail foot, the valid multiplication description also applies here. Since the line support is a quantity that is of little value, the authors divided the value by the standard width of the sleeper in one step ($b_1$). This gives the value of the standard compressive stress acting on the ballast, according to Equation (25).

$$\sigma_z = \Phi \cdot \gamma_r \cdot \frac{R_z}{b_1} = 2.00 \cdot 1.35 \cdot \frac{R_z}{250} = 2.00 \cdot 1.35 \cdot \frac{43.80}{250} = 0.47 \, {}^{N}/_{mm^2} \tag{25}$$

**Remark 3.** *The UIC recommendation [61] omits the line support force calculation issue, so the used multiplication is a unique concept, not a mature solution based on professional consensus.*

On page 136 of reference [55] there is a recommendation of the OSJD (Organization for Cooperation of Railways) for the value of 'bedding' stresses (i.e., the contact pressure

acting onto the railway ballast bed from the sleepers). According to it, no stress higher than $5.0 \, \text{kp/cm}^2$ should be generated under the effect of dynamic loading, which is $0.5 \, \text{N/mm}^2$. The performed calculation resulted in a smaller value, so it is possible to use the prototype concrete sleeper.

**Remark 4.** *It can be seen from the executed calculations that the lower support stiffness has a positive effect on the value of the pressure on the ballast particles. This can be achieved by using under sleeper pads (USPs). Another advantage of the under sleeper pads is that they increase the ballast particles' cross-contact area and reduce the stress on the particle's edges/corners.*

*3.2. Numerical Analysis of the Designed Pre-Stressed Concrete Railway Sleepers Using ABAQUS Finite Element Software*

In this section, the authors introduce their results on detailed FE modeling executed by ABAQUS sophisticated 3D FE software. The numerical simulation aimed to determine more accurate stresses in 3D in the designed railway sleeper, not only in the concrete but in the steel bars. The 3D modeling also allows for the showing of the cross-sectional stress distribution in the mentioned structural elements. These results are adequate for considering them during a detailed design (planning) procedure that takes into account the local stress peaks. To be able to introduce this modeling and its results, the authors need to introduce the basis of the different considerable mechanical models.

3.2.1. Concrete Damage Plasticity Constitutive Model

In the accessible research and books, full descriptions of this model can be found. At this point, a short overview with some additions of the concrete constitutive model is required. Through the usage of the Prandtl-Reuss concept in association with the elasto-plastic deformations, the general strain tensor value $\epsilon_{ij}$ might be divided into an elastic value $(\epsilon_{ij}^{el})$ and a plastic value $(\epsilon_{ij}^{pl})$, as elucidated in Equation (26).

$$\epsilon_{ij} = \epsilon_{ij}^{el} + \epsilon_{ij}^{pl} \tag{26}$$

Also, internal force-strain relations are specified by the elastic damaged scalar equation (see Equation (27)).

$$\hat{\sigma}_{ij} = D_{ijkl}^{el} * (\epsilon_{ij} - \epsilon_{ij}^{pl}) \tag{27}$$

Hereafter, $D_{ijkl}^{el}$ denotes the degraded elastic stiffness, as shown in Equation (28),

$$D_{ijkl}^{el} = (1 - d)D_0^{el} \tag{28}$$

$D_0^{el}$ is the initial (undamaged) elastic stiffness of the material; whereas $d$ denotes the scalar stiffness degradation variable that is extending from zero (the undamaged state of material) to one (the fully damaged state of material). In the theory of the scalar damage, the stiffness drop is isotropic and considered by the variable of degradation $d$. By employing the common concepts of continuum damage mechanics, the actual internal force is outlined in Equation (29).

$$\overline{\sigma}_{ij} = D_0^{el} * (\epsilon_{ij} - \epsilon_{ij}^{pl}) \tag{29}$$

The internal force $\hat{\sigma}_{ij}$ is linked to the actual internal force $\overline{\sigma}_{ij}$ using the scalar reduction relationship (see. Equation (30)).

$$\hat{\sigma}_{ij} = (1 - d) \cdot \overline{\sigma}_{ij} \tag{30}$$

As $d = 0$ when there is no damage, the actual internal force $\overline{\sigma}_{ij}$ would be equivalent to the internal force $\hat{\sigma}_{ij}$. However, once damage takes place, the actual internal force turns out to be more representative if compared with the internal force, such as the external loads

that are resisted by the area of the actual internal force. Equation (27) could be rewritten considering the nominal stress and the reduced elastic tensor offered in Equation (29) constructing Equation (31).

$$\hat{\sigma}_{ij} = (1-d)D_0^{el} \cdot \left(\epsilon_{ij} - \epsilon_{ij}^{pl}\right) \tag{31}$$

The following internal force-strain relationship (see Equation (32)) is creating the damage plasticity constitutive model:

$$\hat{\sigma}_{ij} = (1-d) \cdot \overline{\sigma}_{ij} \rightarrow \hat{\sigma}_{ij} = (1-d_t)\overline{\sigma}_{t_{ij}} + (1-d_c)\overline{\sigma}_{C_{ij}} \tag{32}$$

The compression $d_c$ and tension $d_t$ damage variables are going from representing the undamaged case into 1 representing the fully damaged case, and $\overline{\sigma}_t$ and $\overline{\sigma}_c$ are the tension and compression actual internal force, correspondingly. Commonly, the model used to define the damage in concrete considers the failure method of crushing by compression and cracking in tension. However, the uniaxial reaction of concrete is examined due to the degradation mechanism complication of concrete's uniaxial cyclic behavior by opening and closing the formed micro-cracks. The plasticity damage is expected to affect concrete's uniaxial compressive and tensile reaction, as revealed in Figure 22.

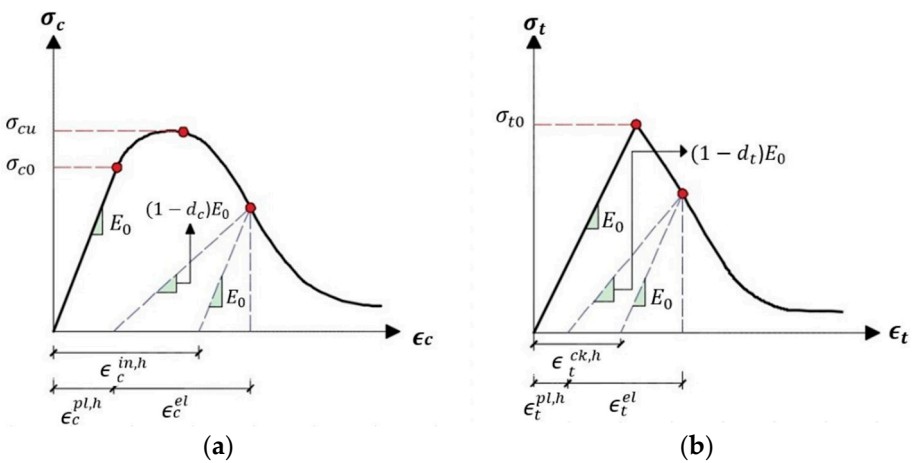

(**a**)                          (**b**)

**Figure 22.** Concrete reaction to uniaxial loading state in: (**a**) compression, (**b**) tension.

The uniaxial compressive and tensile reaction of concrete regarding the concrete damage plasticity model in compression and tension loading is characterized by Equations (33) and (34).

$$\sigma_t = (1-d_t)E_0\left(\epsilon_t - \epsilon_t^{pl,h}\right) \tag{33}$$

$$\sigma_c = (1-d_c)E_0\left(\epsilon_c - \epsilon_c^{pl,h}\right) \tag{34}$$

Presenting $E_0$ as the initial material (undamaged) Young's modulus, whilst $\epsilon_t^{pl,h}$ and $\epsilon_c^{pl,h}$ are the corresponding plastic strains in tension and compression, respectively. Accordingly, the actual uniaxial compressive and tensile stresses $\overline{\sigma}_t$ and $\overline{\sigma}_c$ are presented as shown in Equations (35) and (36):

$$\overline{\sigma}_t = \frac{\sigma_t}{(1-d_t)} = E_0\left(\epsilon_t - \epsilon_t^{pl,h}\right) \tag{35}$$

$$\overline{\sigma}_c = \frac{\sigma_C}{(1-d_c)} = E_0\left(\epsilon_c - \epsilon_c^{pl,h}\right) \tag{36}$$

hence tensile strain $\epsilon_t$ defined by $\epsilon_t^{pl,h} + \epsilon_t^{el}$, and compressive strain $\epsilon_c$ defined by $\epsilon_c^{pl,h} + \epsilon_c^{el}$. Consequently, $\epsilon_t^{el}$ and $\epsilon_c^{el}$ are the corresponding elastic strains in tension and compression, respectively.

### 3.2.2. Details of the Pre-Stressed Sleeper

The sleepers in this research were designed with a length of 260 cm and a variable cross-sectional area. The concrete used to construct the sleepers has the commonly used C50/60 concrete grade properties, while the steel used in reinforcing the sleepers is defined as a high-strength steel (see Section 3.2.3). Figure 1 shows the geometric dimensions of the sleeper specimen, while the reinforcement details are shown in Figure 23, where 11 pieces of φ6 mm stirrups were each distributed along 25 cm with the sleeper in order to prevent the appearance of the shear cracks, while the sleeper was reinforced longitudinally by 12 pieces of φ6 mm pre-stressed bars to resist the tensile forces.

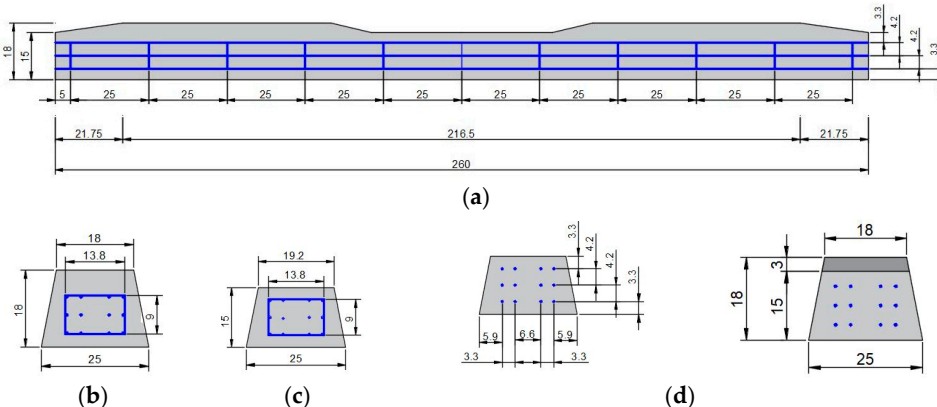

**Figure 23.** Reinforcement details with: (**a**) Side view, (**b**) Section under the rail, (**c**) Section at mid-span, (**d**) End section. (The dimensions are given in centimeters).

### 3.2.3. Numerical Modeling of the Sleeper by ABAQUS FE Software

Basically, ABAQUS software was used with the help of the Concrete Damage Plasticity (CDP) model, which is used to calibrate the concrete behavior in both tension and compression states in order to model the sleeper specimens numerically. As CDP is presumed to describe the concrete response in tension and compression, it was possible to acquire the curves demonstrated in Figure 24, which display the nonlinear behavior of concrete, in both the tension and compression states.

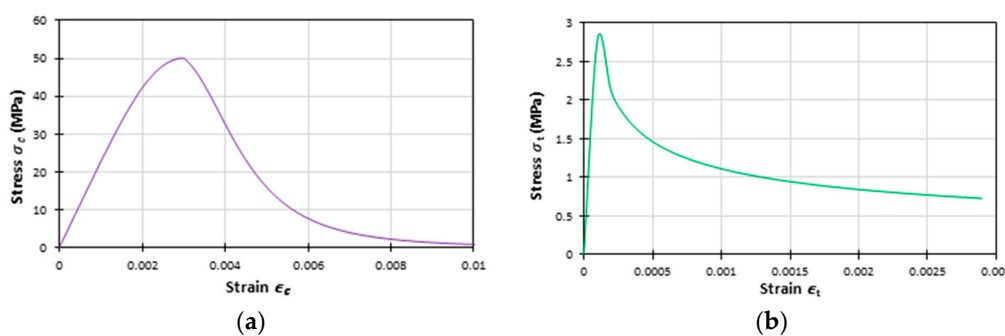

**Figure 24.** Concrete properties (responses) in (**a**) compression and (**b**) tension.

Furthermore, these properties are introduced inside ABAQUS to acquire CDP parameters that imitate the required damage behavior of concrete; after applying sensitivity analyses, the inputted CDP parameters are assumed as given in Table 10, where in the FEM analysis, Young's modulus of concrete was quantified as $E_0$ = 37,300 MPa, and the Poisson's ratio is $\nu$ = 0.2.

**Table 10.** CDP input data for concrete.

| Dilation Angle | Eccentricity | $f_{b0}/f_{c0}$ | K |
|:---:|:---:|:---:|:---:|
| 31 | 0.2 | 1.16 | 0.667 |

As mentioned previously, the sleepers were reinforced using pre-stressing tendons; these tendons that are considered in this study possess a yield strength of 1700 MPa, a Modulus of Elasticity of 210,000 MPa, and a Poisson's ratio of $\nu = 0.3$, while the pre-stressed applied loading inside the tendons is assumed to be equal to 1400 MPa. In order to model the pre-stressing effect in ABAQUS, a pre-defined temperature-load is applied for the defined tendons. Since mesh size affects the results accuracy as well as the simulation time, a size study was considered to check the effect of mesh size on accuracy and calculation time. An optimum mesh size of 25 mm was then applied to obtain an accurate result, where the total element number was approximately equaled to 15,500 elements, as shown in Figure 25, as a ten-node general-purpose quadratic was employed to define concrete material, and two-node 3D truss elements defined the steel material, where steel bars interacted with concrete using the embedded region option. Furthermore, in addition to the initial step that contains the creation of boundary conditions, two more steps were considered in the numerical analysis whereby the first step was applied to present the pre-stressed (pre-defined temperature) effect, while the other step had the task of analyzing and computing the outputs by applying the Static Risk (SR) concentrated force. As a result, the numerical models were obtained as shown in Figure 26.

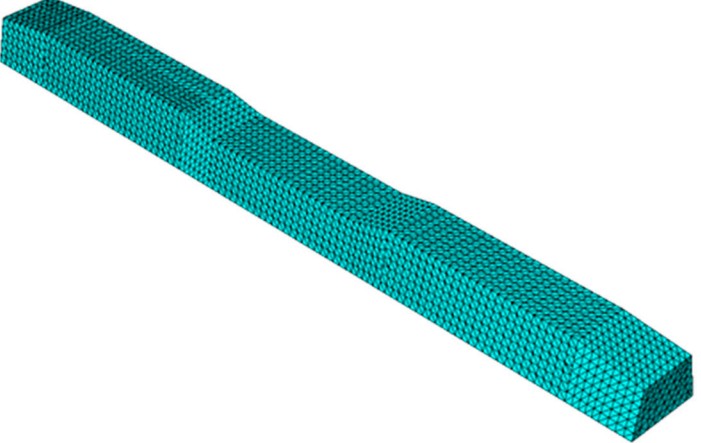

**Figure 25.** Meshing of the numerical model in ABAQUS.

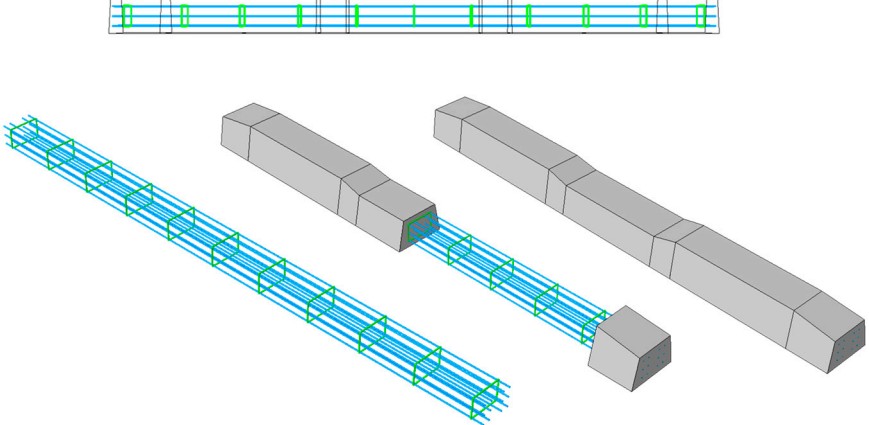

**Figure 26.** Numerical model of the concrete sleeper in ABAQUS.

In this research, two test methods were established; in the first one, the sleeper was under a one-point load which was considered at the mid-span of the sleeper. In the second method, the load was considered on the base plate, which is connected to the rail of the railway where Figure 27 shows both loading circumstances and supporting details. Both loadings and boundary conditions were chosen to match following EN 13230-1 [64] and EN 13230-2 [65] standards. For this reason, the sleepers were simply hinge-supported. In the numerical work, the concentrated load was considered vertically to the surface of loading, by one point, and considering the couple-effect upon a defined area. The standard values of $L_r$ and $L_c$ were taken into consideration as 0.6 m and 1.5 m, according to EN 13230-2 [65]. $L_r$ and $L_c$ are the bay length of the sleeper during the bending tests.

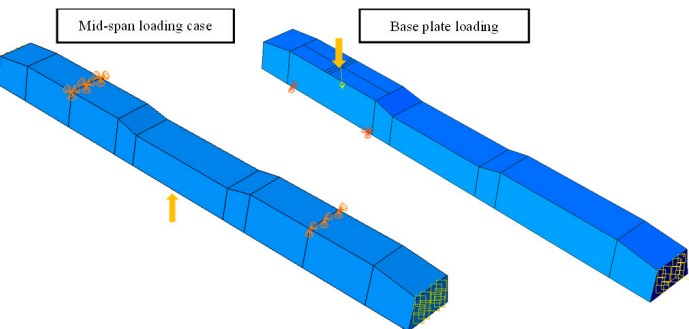

**Figure 27.** Loading conditions and supporting details related to the running in ABAQUS. (The mid-span loading case is rotated 180° in reality; this figure is only for comparison and understanding).

After completing the modeling process, the results were obtained and plotted for comparison, as presented in Figure 28, demonstrating the $P$–$\Delta$ response for both models owning different loading conditions; where $P$ is the vertical force (load) in the kN unit, and $\Delta$ is the vertical deflection of the sleeper in mm units. By observing the curves, the positive moment test, considering mid-span load, produced a lower load value when compared with the other test type, as more than half this value reduced the loading value, and it can be explained by knowing that when the $a/d$ ratio decreases, the ultimate load value produced would be higher. Moreover, it can be seen that the mid-span loading case gave higher deflection values if compared with the other case, as in this case a higher bay length is applied, and the increased length allowed the sleeper to act in a more elastic way, which reflected the corresponding deflection values.

Additionally, more results were obtained to understand both specimens' general behaviors; Figures 29 and 30 show different result types for the sleepers loaded in mid-span and in the base plate, respectively. Apparently, the damaged parts are located in the region between the supports where the highest stresses are initiated, and Figure 29a reveals that the top part that is colored red is the most damaged part in the concrete, indicating that the intensity of damage is specified by colors, as red reflects the fully damaged parts ($d_t = 1$), while blue colors reflect the undamaged parts ($d_t = 0$). It is worth mentioning that in this case, the damage was located at the top part of the sleeper as a consequence of applying the load in the upward direction, which caused the upper fibers to undergo high tensile stresses, and this consequently caused the accumulation of the damaged parts there. In addition, Figure 29a shows that steel tendons undergo a yielding state at the specified failure area—between supports—defining higher stress intensity (normal stress $\sigma$-to-yield stress $\sigma_y$ ratio) with red color, which gradated, reaching a blue color that represents lower stress intensity in steel.

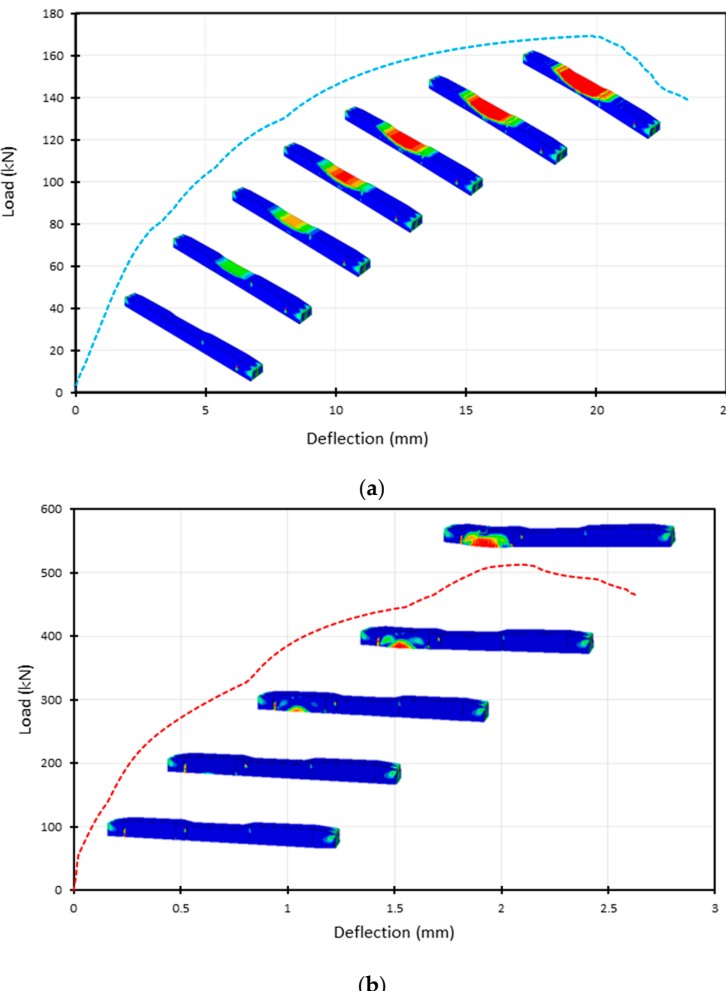

**Figure 28.** *P–Δ* response for sleepers with different loading conditions based on the running with ABAQUS: (**a**) Mid-span loading case, (**b**) Base plate loading. (The deflection parameter in the horizontal axis is the max. vertical deformation during loading).

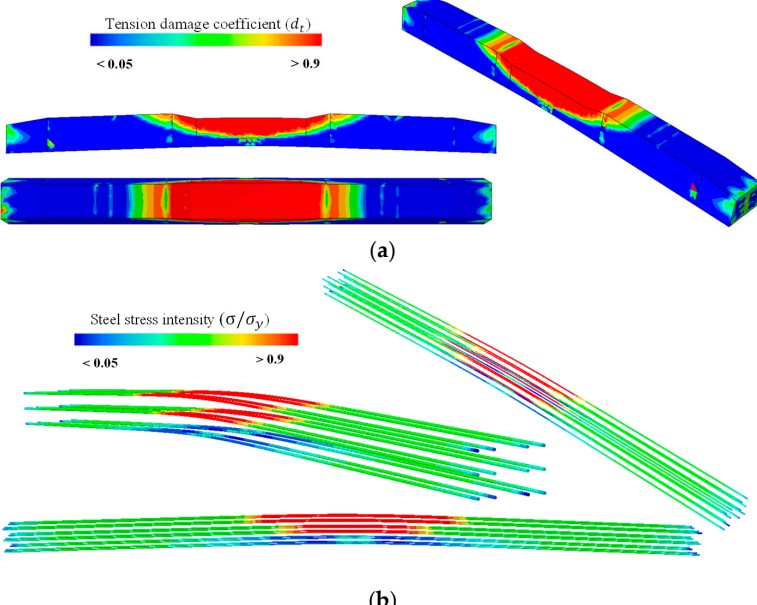

**Figure 29.** Mid-span loading case numerical results based on the running with ABAQUS. (**a**) Tension damage ($d_t$); (**b**) steel stress intensity ($\sigma/\sigma_y$).

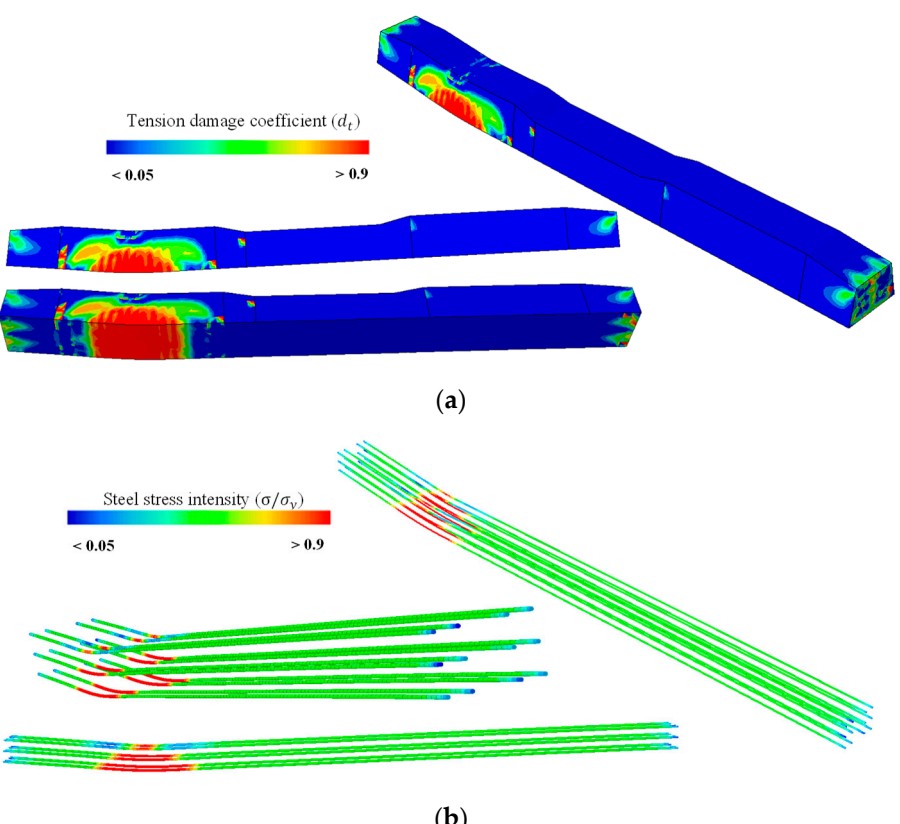

**(a)**

**(b)**

**Figure 30.** Base plate loading case numerical results based on the running with ABAQUS.

On the other hand, Figure 30a shows narrower affected and damaged parts due to the small distance between the supports, which also caused an almost total coverage of the damage to that small area; however, larger loads were achieved. The damaged red areas are located at the lower part as a result of applying the loading in the downward direction, causing a negative moment condition. Similarly, Figure 30b displays the steel stress intensity, revealing that tendons undergo the higher stresses in the part between the supports, while the lower tendons get greater damage if compared with the other layers.

The authors would like to show a more detailed picture and exact behavior of the sleepers, as well as the stress distribution in its parts related to both loading cases; this is why they prepared cross-sectional and longitudinal-sectional diagrams and figures with regard to them.

First, the mid-span loading case is taken into account. Then, different sections were considered, as clarified in Figure 31, where section A-A divide the sleeper into two halves in the width direction, B-B sections cuts across the sleeper section at the loading point, and the C-C section cuts through the support line.

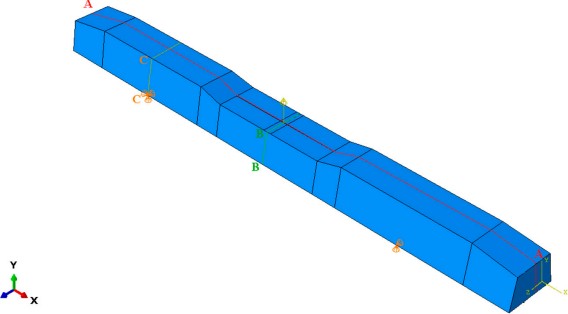

**Figure 31.** Considered sections related to a mid-span loading case in ABAQUS.

Figures 32–34 show the concrete tension damage, stress values inside the longitudinal steel, and vertical deflection value, where these values were considered at the A-A section.

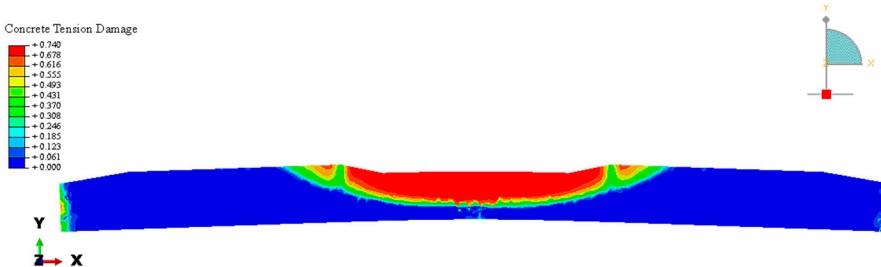

**Figure 32.** Concrete tension damage results in longitudinal section A-A related to a mid-span loading case in ABAQUS.

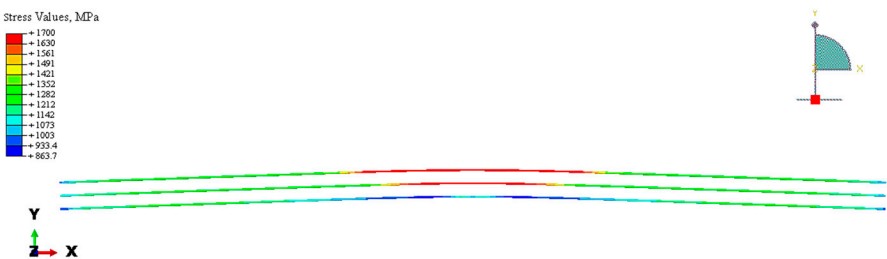

**Figure 33.** Stress results of steel bars in longitudinal section A-A related to mid-span loading case in ABAQUS.

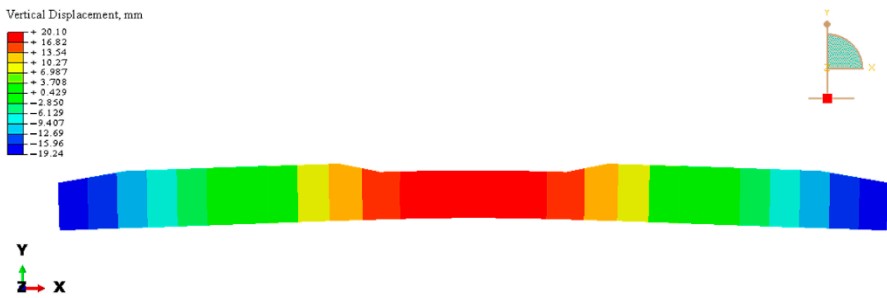

**Figure 34.** Vertical displacement results of steel bars in longitudinal section A-A related to a mid-span loading case in ABAQUS.

Moreover, Figure 35 illustrates the concrete tension damage at the B-B section (under loading point), where the damage can be at its maximum.

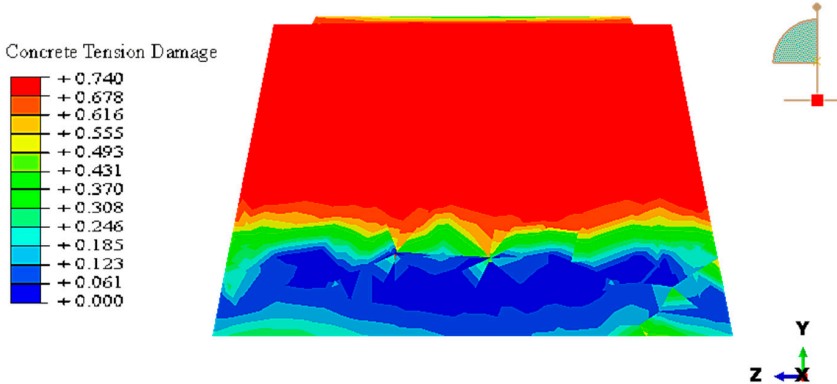

**Figure 35.** Concrete tension damage results in cross-section B-B related to a mid-span loading case in ABAQUS.

Figures 36–38 show the stress values for each steel bar in the supporting section (C-C section; see Figure 38) and in the loading section (B-B section; see Figures 36 and 37). It can be seen that steel bars had small stress values in the support area and are not yielded.

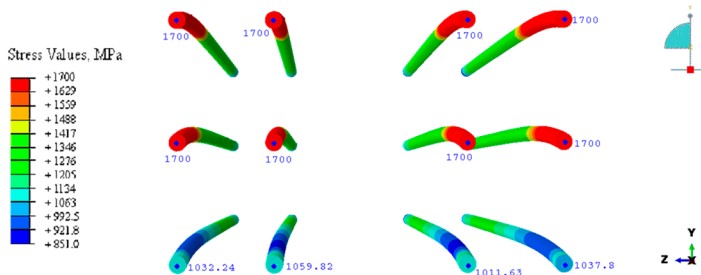

**Figure 36.** Stress results (1) of steel bars in cross section B-B related to a mid-span loading case in ABAQUS.

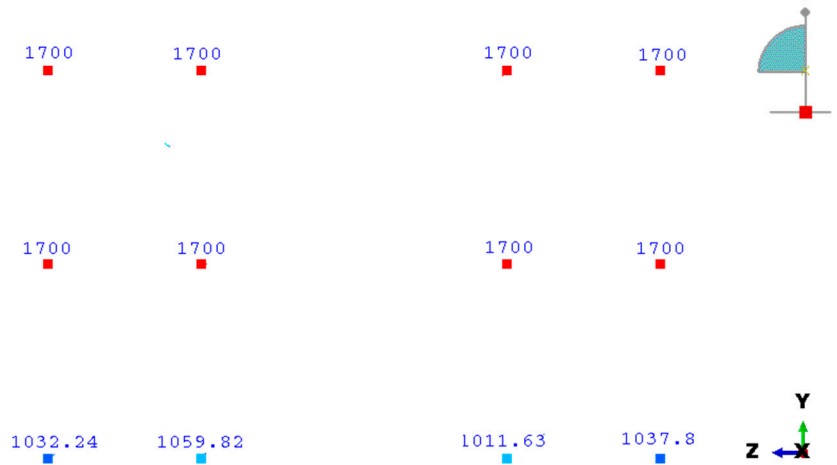

**Figure 37.** Stress results (2) of steel bars in cross section B-B related to a mid-span loading case in ABAQUS.

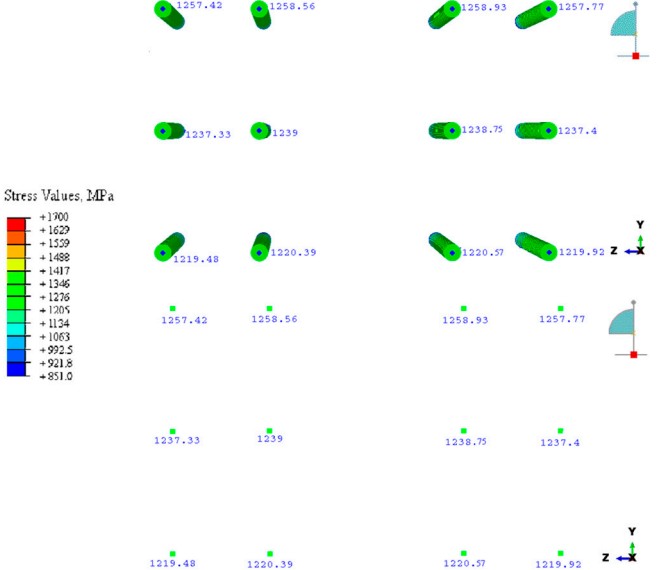

**Figure 38.** Stress results of steel bars in cross section C-C related to a mid-span loading case in ABAQUS.

Additional sections were provided to create a clearer view of the results obtained. In Figure 39, it can be seen that section A-A was considered to divide the sleeper into

two equal parts in the width direction, while section B-B provides a cut under the loading point, while sections C-C and D-D represent a cross section in the outer and inner supports, respectively.

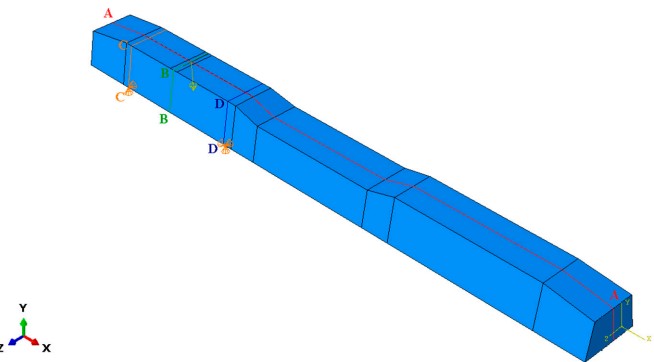

**Figure 39.** Considered sections related to a base plate loading case in ABAQUS.

Figures 40–42 were produced by the A-A section providing concrete tension damage, steel stresses, and vertical deflection values.

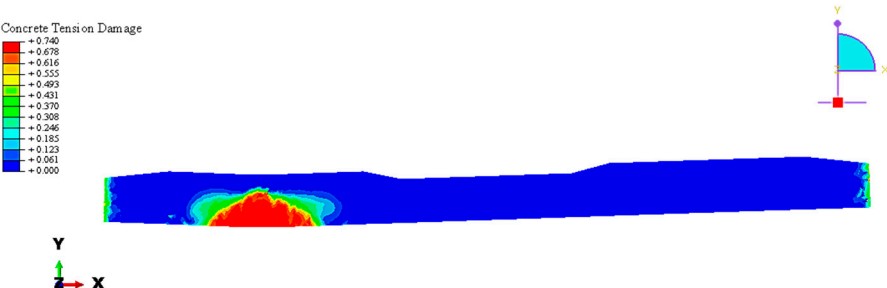

**Figure 40.** Concrete tension damage results in longitudinal section A-A related to a base plate loading case in ABAQUS.

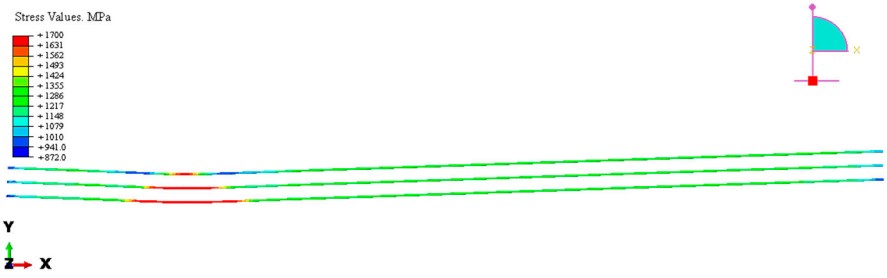

**Figure 41.** Stress results of steel bars in longitudinal section A-A related to a base plate loading case in ABAQUS.

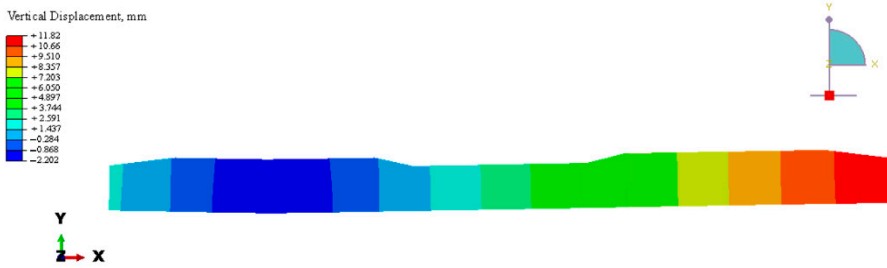

**Figure 42.** Vertical displacement results of steel bars in longitudinal section A-A related to a base plate loading case in ABAQUS.

Figure 43 illustrates the concrete tension damage at the B-B section (under loading point), where the damage can be at its maximum.

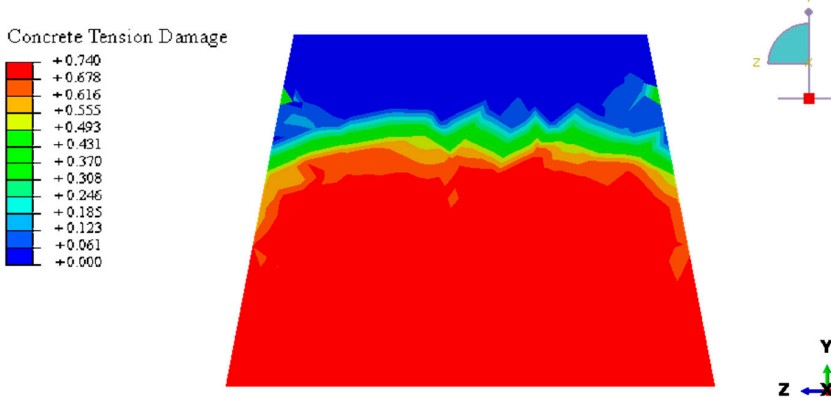

**Figure 43.** Concrete tension damage results in a B-B related cross-section to the base plate loading case in ABAQUS.

Additionally, Figures 44–46 show the stress values for each steel bar in the loading section (B-B section) and in the supporting sections (C-C and D-D). It can be seen that steel bars had small stress values in the support areas, and they are not yielded.

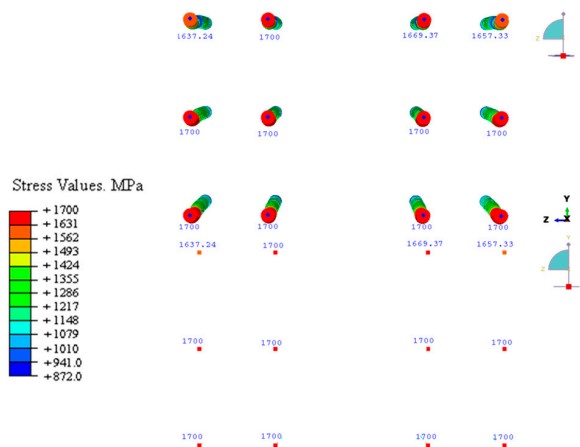

**Figure 44.** Stress results of steel bars in a B-B related cross section to the base plate loading case in ABAQUS.

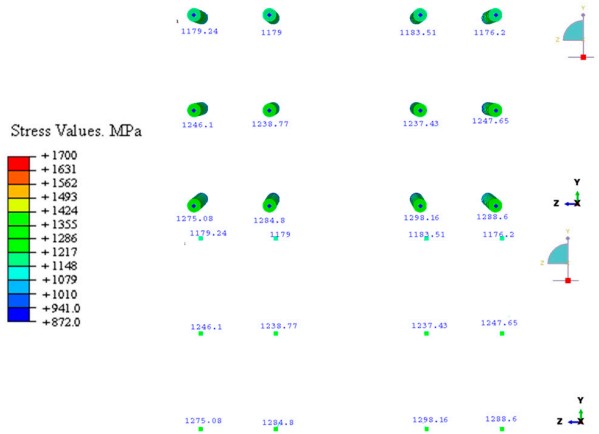

**Figure 45.** Stress results of steel bars in a C-C related cross section to the base plate loading case in ABAQUS.

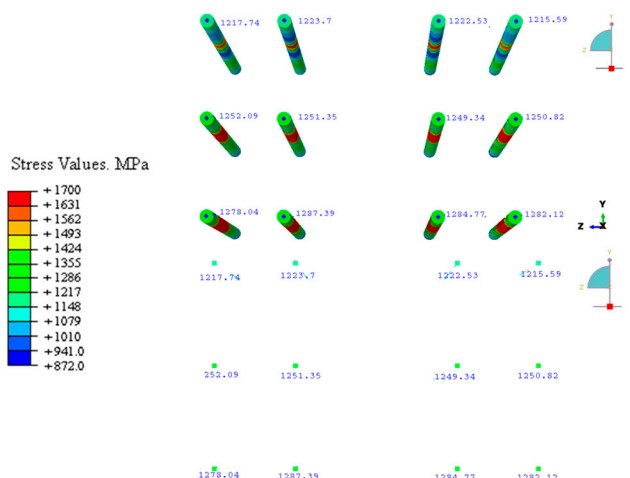

**Figure 46.** Stress results of steel bars in a D-D related cross section to the base plate loading case in ABAQUS.

## 4. Discussion

The authors showed the obtained results in Section 3.1 related to the finite element modeling using AxisVM software; in addition, Section 3.2 contains the results regarding finite element modeling with the help of ABAQUS. It has to be mentioned that in Section 3.1 the considered loading and the load arrangement mainly focus onto the design and the so-called dimensioning process; however, Section 3.2 provides details about the standardized laboratory test arrangement and the loading of the previously designed reinforced concrete railway sleeper with the aid of the sophisticated FE software, ABAQUS. The detailed modeling means that ABAQUS is able to take into consideration the stress-strain state of the structures with the almost real material models and their behavior.

The authors wanted to show the main differences between the FE modeling and the received results of the considered RC sleeper. A simple, base FE software at this field with the base functions cannot provide the ability to model the structure as a real 3D element applying the 3D mesh, and the loading procedure with accurate material models also cannot be used during the running. The AxisVM is a so-called 2D or 2.5D software which allows calculations in 2D with the possibility of the consideration of the width or thickness of the structure, as well as the visualization and the results, which are plotted in 3D figures. In some cases, so-called shell models are able to be applied, but these models cannot consider the real 3D behavior of the engineering structures. This is the reason why this methodology is not appropriate for the calculation of the stress-strain state in 3D. There are two standardized laboratory test arrangements based on EN 13230-1 [64] and EN 13230-2 [65] standards.

The authors presented a solution for determining the inner forces of a designed pre-stressed reinforced concrete railway sleeper for a high-speed railway application based on finite element modeling (with AxisVM software) and UIC 713 [59]. It can be seen that the FE method correlates extremely well with the calculation according to UIC 713 [59] for bending moment values. Furthermore, it complements the points in the recommendation [59] when determining the shear stress and pressure (normal stress) on the ballast bed, as the UIC 713 [59] neglects these parameters. The advantage of the alternative method is that in case of changing the parameters within reasonable limits, the designed sleeper can be used without a separate test and analysis in the case of applying under sleeper pads (USPs).

In the second part of the paper (see Section 4), the authors introduced a methodology for more precise FE modeling of the designed railway sleeper executed in the first part (see Section 3). The basis of the modeling was the standard laboratory tests of the pre-stressed concrete railway sleepers (see EN 13230-1 [64] and EN 13230-2 [65]). The 3D finite element modeling was able to help with the determination of local stresses, as well as local and

overall deformations of the elements and the whole sleeper. The cracks and crackings were not considered during the runnings, only the introduced material model was.

The results show that the max. deflection values and max. vertical forces were approximately 20 mm and 170 kN,;as well as 2 mm and 510 kN, related to the mid-span and base-plate loadings, respectively. They indicate approximately 63.75 kNm ($M_{dc-,breakage}$) and 76.5 kNm ($M_{dr+,breakage}$) values. To be able to compare the results with the FE modeling based on AxisVM, Table 7 is important. The modeling of ABAQUS does not consider the prestressing loss, which can be approximately 40%. If this loss value was taken into consideration, the modified results would be 38.25 and 45.9 kNm (while the initial prestressing of the tendons was 1400 MPa) related to the $M_{dc-,breakage}$ and $M_{dr+,breakage}$, respectively. The difference between the results from the FE modeling of AxisVM and from ABAQUS is 13.67% in the case of base plate loading because it is the standard (critical) case.

The authors certified that the methodology of FE simulation is appropriate to determine the extreme values of the inner forces and the stresses of the pre-stressed reinforced concrete railway sleepers. The AxisVM software is mainly adequate for determining inner forces; however, the ABAQUS FE software is suitable for applying sophisticated material models.

## 5. Conclusions

The authors dealt with the finite element modeling of a reinforced concrete railway sleeper designed with detailed dimensioning appropriate for a design speed of 300 km/h and a design axle load of 180 kN.

In the first part of the paper, the dimensioning methods, the determined sizes, and the parameters were introduced and shown. The dimensioning process was aided by the FE modeling with the AxisVM software. It helped with the determination of the inner forces of the structure with the consideration of different support conditions.

After obtaining the sizes, dimensions, and types of materials, the ABAQUS sophisticated FE software was applied to take into consideration two laboratory test arrangements based on EN 13230-1 [64] and EN 13230-2 [65] standards. In addition, a special elastoplastic concrete material model was applied during the modeling. Finally, the detailed stress-strain results of the elements (concrete, steel tendons, etc.) were introduced and shown. The production technology of the designed sleeper can be a standard long-line system method.

The authors certified that the methodology of FE simulation is appropriate to determine the extreme values of the inner forces and the stresses of the pre-stressed reinforced concrete railway sleepers. The AxisVM software is mainly adequate for determining inner forces; however, the ABAQUS FE software is suitable for applying sophisticated material models.

The model can be developed in the future with the following aspects:

- considering real laboratory tests and validating the results from FE modeling with them;
- considering material models that allow for the calculation of crackings and their effect in the concrete, in addition to being based on measurements of real cubes and cylindrical specimens in the laboratory (compressive and splitting tensile tests);
- considering more sophisticated models with regard to ballast behavior [66] and/or dynamic stress analysis [67];
- the real pattern of the crackings can be measured by GOM DIC technology [11,68–75];
- taking into consideration the prestressing loss more exactly instead of the approximate calculation;
- the ABAQUS model and the traditional (hand-made) calculation can be compared.

The experiences can be adequate for the of dimensioning of other (modified) sleeper geometries and applications (narrow and wide gauges; higher axle loads), as well as for support geometries/patterns.

**Author Contributions:** Conceptualization, Z.M., M.S. and D.K.; methodology, Z.M., M.M.R., M.S., D.K. and S.F.; software, Z.M, S.K.I. and M.M.R.; validation, Z.M., M.M.R., M.S., D.K. and S.F.; formal analysis, S.K.I. and M.M.R.; investigation, Z.M., S.K.I., M.M.R., A.N., D.H.(Dániel Harrach), G.H., S.S., S.K.S., D.H. (Dóra Harangozó), M.S., D.K., G.B., L.G. and S.F., resources, S.F.; data curation, Z.M., S.K.I., M.M.R., A.N., D.H. (Dániel Harrach), G.H., S.S., S.K.S., D.H. (Dóra Harangozó), M.S., D.K., G.B., L.G. and S.F.; writing—original draft preparation, Z.M., S.K.I., M.M.R., A.N., D.H. (Dániel Harrach), G.H., S.S., S.K.S., D.H. (Dóra Harangozó), M.S., D.K., G.B., L.G. and S.F.; writing—review and editing, Z.M., S.K.I., M.M.R., A.N., D.H. (Dániel Harrach), G.H., S.S., S.K.S., D.H. (Dóra Harangozó), M.S., D.K., G.B., L.G. and S.F.; visualization, Z.M, S.K.I., M.M.R. and S.F.; supervision, M.M.R., A.N., S.S., S.K.S., M.S., D.K., L.G. and S.F.; project administration, S.F.; funding acquisition, S.F. All authors have read and agreed to the published version of the manuscript.

**Funding:** This research received no external funding.

**Data Availability Statement:** Not applicable.

**Acknowledgments:** This paper was prepared by the research team "SZE-RAIL".

**Conflicts of Interest:** The authors declare no conflict of interest.

## Abbreviations/Nomenclature

**Abbreviations**

| | |
|---|---|
| 2D | two dimensions or two dimensional |
| 2.5D | two and a half dimensions or two and a half dimensional |
| 3D | three dimensions or three dimensional |
| ABAQUS | a sophisticated software based on finite element method calculation which is mainly used for research and development |
| AxisVM | a software based on finite element method calculation established in Hungary that is mainly applied for engineering design |
| CDP | Concrete Damage Plasticity |
| DE | discrete element |
| DEM | discrete element method or discrete element modeling |
| DIC | digital image correlation |
| FE | finite element |
| FEM | finite element method or finite element modeling |
| L-CFRPU | laminated carbon fiber reinforced polyurethane |
| MSFRC | macro synthetic fiber-reinforced concrete |
| POSTSRC | post-stressed reinforced concrete |
| PRESRC | pre-stressed reinforced concrete |
| RC | reinforced concrete |
| RPC | reactive powder concrete |
| SR | static risk |
| UHPC | ultra-high performance concrete |
| UHP-FRC | ultra-high performance fiber-reinforced concrete |
| USP | under sleeper pad |

**Nomenclature**

| | |
|---|---|
| $\nu$ | Poisson's ratio [$-$] |
| $\rho$ | Density [kg/m$^3$] |
| $\Gamma$ | The multiply of $\gamma_r$ and $\gamma_i$, introduced factor [$-$] |
| $\varphi$ | Speed factor according to the Eisenmann equation [$-$] |
| $\Phi$ | Dynamic factor according to Eisenmann equation [$-$] |
| $\sigma$ | Normal stress [MPa] or [Pa] |
| $\sigma_c$ | Compression (compressive) stress [MPa] or [Pa] |
| $\sigma_t$ | Tensile stress [MPa] or [Pa] |
| $\sigma_y$ | Yield stress [MPa] or [Pa] |
| $\bar{\sigma}_{Cij}$ | Actual internal compression force [MPa] or [Pa] |
| $\hat{\sigma}_{ij}$ | Internal force tensor value [MPa] or [Pa] |
| $\bar{\sigma}_{ij}$ | Actual internal force [MPa] or [Pa] |

| | | |
|---|---|---|
| $\overline{\sigma}_{tij}$ | Actual internal tension force [MPa] or [Pa] | |
| $\Delta$ | Vertical deflection of the sleeper in ABAQUS modeling [mm] | |
| $\varepsilon_c$ | Compression strain [$-$] | |
| $\varepsilon_{ij}$ | General strain tensor value [$-$] | |
| $\varepsilon_{ij}{}^{el}$ | Elastic part of the general strain tensor value [$-$] | |
| $\varepsilon_{ij}{}^{pl}$ | Plastic part of the general strain tensor value [$-$] | |
| $\varepsilon_t$ | Tensile strain [$-$] | |
| $\epsilon_c^{el}$ | Elastic strain in compression [$-$] | |
| $\epsilon_c^{pl,h}$ | Plastic strain in compression [$-$] | |
| $\epsilon_t^{el}$ | Elastic strain in tension [$-$] | |
| $\epsilon_t^{pl,h}$ | Plastic strain in tension [$-$] | |
| $A$ | Supporting contact area regarding one rail [mm$^2$] | |
| $A_1$ | Shear area associated with shear forces in local 1st direction [cm$^2$] | |
| $A_2$ | Shear area associated with shear forces in local 2nd direction [cm$^2$] | |
| $A_x$ | Axial (cross-sectional) area [cm$^2$] | |
| $A_y$ | Shear area in local $y$ direction [cm$^2$] | |
| $A_z$ | Shear area in local z direction [cm$^2$] | |
| $b_1$ | Width of sleeper bottom [mm] | |
| $C$ | Bedding modulus below the sleeper's bottom [N/mm$^3$] | |
| $d$ | Thickness of the sleeper under the rail foot's center [m] | |
| $d'$ | Scalar stiffness degradation variable [$-$] | |
| $d_c$ | Compression damage variable [$-$] | |
| $d_t$ | Tension damage variable [$-$] | |
| $D_0^{el}$ | Initial (undamaged) elastic stiffness of the material [kN/mm] | |
| $D_{ijkl}^{el}$ | Degraded elastic stiffness [kN/mm] | |
| $e$ | Random eccentricity [m] or [mm] | |
| $E_0$ | Material initial (undamaged) Young's modulus | |
| $E_x$ | Young's modulus of elasticity in the local $x$ direction [kN/cm$^2$] | |
| $E_y$ | Young's modulus of elasticity in the local $y$ direction [kN/cm$^2$] | |
| $f$ | Half of the load distribution width related to the sleeper [m] | |
| $f_{b0}$ | Equibiaxial compressive strength of concrete [MPa] | |
| $f_{c0}$ | Uniaxial compressive strength of concrete [MPa] | |
| $F_z$ | Maximal vertical forces acting onto the sleepers [kN] | |
| $I_1$ | Principal inertia about local 1st axis [cm$^4$] | |
| $I_2$ | Principal inertia about local 2nd axis [cm$^4$] | |
| $I_x$ | Torsional inertia [cm$^4$] | |
| $I_y$ | Flexural inertia about local $y$ axis [cm$^4$] | |
| $i_y$ | Radius of inertia about $y$ axis [cm] | |
| $I_{yz}$ | Centrifugal inertia [cm$^4$] | |
| $I_z$ | Flexural inertia about local $z$ axis [cm$^4$] | |
| $i_z$ | Radius of inertia about $z$ axis [cm] | |
| $I_\omega$ | Warping modulus [cm$^6$] | |
| $K$ | Ratio of the second stress invariant on the tensile meridian in ABAQUS modeling [$-$] | |
| $K_z$ | Vertical spring constant for one rail [kN/m] | |
| $L_c$ | Design distance between center lines of the rail seat for the test arrangement of mid-span loading based on EN 13230-2 [62] [m] | |
| $L_r$ | Design distance between the articulated supports center lines for the test arrangement at the rail seat section based on EN 13230-2 [62] [m] | |
| $L_S$ | Length of the sleeper [mm] | |
| $M$ | Mean value of bending moment [kNm] | |
| $M_d$ | Design value of bending moment [kNm] | |
| $M_{dc-}$ | Negative extreme value of the bending moment causing cracking related to the sleeper's center [kNm] | |

| | |
|---|---|
| $M_{dc+}$ | Positive extreme value of the bending moment causing cracking related to the sleeper's center [kNm] |
| $M_{dr-}$ | Negative extreme value of the bending moment causing cracking under the rail foot [kNm] |
| $M_{dr+}$ | Positive extreme value of the bending moment causing cracking under the rail foot [kNm] |
| $M_{dc-,breakage}$ | Negative extreme value of the bending moment causing breakage in the sleeper's center [kNm] |
| $M_{dr+,breakage}$ | Positive extreme value of the bending moment causing breakage under the rail foot [kNm] |
| $M_y$ | Bending moment (around the y axis) [kNm] |
| $n$ | Track condition parameter according to the Eisenmann equation [−] |
| $P$ | Vertical load acting on the sleeper in ABAQUS modeling [kN] |
| $P_i$ | Inner circumference (holes) [cm] |
| $P_o$ | Outer circumference (cross-section contour) [cm] |
| $Q_v{'}$ | Specified maximum of the $Q_v$ [kN] |
| $Q_{v1}$ | One of the vertical wheel forces in a wheelset [kN] |
| $q_{v1}$ | Vertical force acting on the one rail [kN] |
| $Q_{v2}$ | The other vertical wheel force in the same wheelset [kN] |
| $q_{v2}$ | Vertical force acting on the other rail [kN] |
| $q_z$ | Linear distributed vertical forces acting onto the sleepers [kN] |
| $r$ | Distance between rail centers, its nominal value is 1.5 m if there is a standard track gauge (1.435 m) |
| $R_z$ | Line support force of the sleeper [kN/m] |
| $s$ | Width of the rail foot [m] |
| $t$ | Student distribution parameter according to the Eisenmann equation [−] |
| $t_R$ | Distance between the track rail axes [mm] |
| $V$ | Allowed speed on the track [km/h] |
| $V_z$ | Shear force of the sleeper [kN] |
| $V_{z,Ed}$ | Design value of the shear force of the sleeper [kN] |
| $W_{y,el,b}$ | Elastic cross-section modulus, regarding to $y$ axis, bottom [cm$^3$] |
| $W_{y,el,t}$ | Elastic cross-section modulus, regarding to $y$ axis, top [cm$^3$] |
| $W_{y,pl}$ | Plastic cross-section modulus regarding to $y$ axis [cm$^3$] |
| $W_{z,el,b}$ | Elastic cross-section modulus, regarding to $z$ axis, bottom [cm$^3$] |
| $W_{z,el,t}$ | Elastic cross-section modulus regarding the $z$ axis, top [cm$^3$] |
| $W_{z,pl}$ | Plastic cross-section modulus regarding the $z$ axis [cm$^3$] |
| $y_G{}^*$ | Position of the center of gravity of the cross-section in local $y$ direction relative to the lower-left corner of the circumscribed rectangle [cm] |
| $y_s$ | Position of the shear center in local $y$ directions relative to the center of gravity [cm] |
| $z_G{}^*$ | Position of the center of gravity of the cross-section in local $z$ direction relative to the lower-left corner of the circumscribed rectangle [cm] |
| $z_s$ | Position of the shear center in local $z$ directions relative to the center of gravity [cm] |
| $\alpha_T$ | Linear thermal expansion ratio [1/°C] |
| $\gamma_d$ | Safety factor due to the load distribution [−] |
| $\gamma_i$ | Safety factor due to the support faults [−] |
| $\gamma_p$ | Safety factor due to the damping effect of the rail pad [−] |
| $\gamma_r$ | Safety factor due to the geometrical faults [−] |
| $\gamma_V$ | Safety factor due to the speed [−] |
| $\alpha$ | Angle between local 1st axis and the local $y$ axis [°] |
| $\rho_1$ | Shear factor for local 1st direction [−] |
| $\rho_2$ | Shear factor for local 2nd direction [−] |
| $\rho_y$ | Shear factor in local $y$ direction [−] |
| $\rho_{yz}$ | Shear factor for local $y$–$z$ cross [−] |
| $\rho_z$ | Shear factor in local $y$ direction [−] |

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
