# Peer review of "Numerical Investigation of Pre-Stressed Reinforced Concrete Railway Sleeper for High-Speed Application"

_infrastructures, doi:10.3390/infrastructures8030041_

Round 1
Reviewer 1 Report
The paper deals with a numerical investigation of pre-stressed reinforced concrete railway sleeper for high-speed application. The considered railway speed is 300 km/h, and the axle load is 180 kN. The authors performed manual and numerical computations and compared the obtained results, thus verifying the used approaches.
The work is original and it is generally well written and structured. Objectives were adequately stated and fulfilled. The quality of presentation is also good. The researchers in this field of work would have interest in this paper.
Certain improvements though still need to be given attention:
1) There is no need and it is actually unusual to structure the abstract the way it has been done. The abstract should have a form of a fluent text and the context reveals which aspect of the paper is addressed in each of the sentence.
2) The authors should discuss the prestressing of tendons in more details, and try to explain why the prestressing loss is neglected in the calculation.
3) The reference list needs to be extended. Regarding the prestress and options for fast determinantion of highly stressed regaions of the structure, this reference may be commented:
Strzalka, C., Marinkovic, D., & Zehn, M. W. (2021). Stress Mode Superposition for a Priori Detection of Highly Stressed Areas: Mode Normalisation and Loading Influence. Journal of Applied and Computational Mechanics, 7(3), 1698-1709. doi: 10.22055/jacm.2021.36637.2878
Another reference to be considered:
Liu, J., Sysyn, M., Liu, Z., Kou, L., Wang, P. Studying the Strengthening Effect of Railway Ballast in the Direct Shear Test due to Insertion of Middle-size Ballast Particles. Journal of Applied and Computational Mechanics, 2022; 8(4): 1387-1397. doi: 10.22055/jacm.2022.40206.3537
Further novel references may be considered as well.
3) Please specify the possible production technology of the designed railway sleeper. This would be of interest to the readers.
4) Most of the figures use decimal commas instead of decimal dots, please make a suitable correction.
5) Ref. 68 is given as an accepted manuscript. The doi number would be a mimimum to be used as bibliographic data.
6) Ref 8 has the number 11420 at the end, which is not correct, as this journal does not have article numbers as identifyers. The doi number is to be used. Also, the journal title in references 6 and 8 is not correct and should read: Facta Universitatis-Series Mechanical Engineering.
7) Can the prestressing loss be considered in the design process and dimensioning of the structure? If yes, what is the proper way of considering this influence.
8) Assuming an increased train speed, of say, 350 km/h, how would this affect the design and dimensions of the structure? Would the procedure be the same?
Author Response
See the attached PDF file. Remark: there is the changes tracked at the end of the file to be able to check them.

Reviewer 2 Report
Dear Authors,
This paper is interesting and useful for further research in the subject field of railway engineering. You performed detailed analyses (analytical and numerical) through clear explanations of all steps of the research. I have a few questions directed at the prototype, material, and experimental results (which are not presented in this paper).
Please consider the following suggestions and questions:
1. Page 5, line 217: Do you have produced a prototype of a concrete sleeper? If you have some images of the prototype, it could be good to insert one of them into the paper.
2. Page 7: Did you perform some tests to get the mechanical and physical properties of the used material (on specimens)? I am not sure which part is simulated with S235.
3. Page 7, line 282: Value of C=0.20 N/mm3 is applied according to the author's experiences? If you have any previous papers which analyzed this parameter, please insert them in this paper (in the ref. list).
4. Page 15, line 450: Obviously is a good correlation between analytical (UIC) and numerical results. Do you have any experimental results obtained by the experimental test of the prototype?
Best regards
Author Response
See the attached PDF file. There is a comparison showing the changes tracked in the manuscript at the end of the file. Please check.

Reviewer 3 Report
1- please door use etc. it is not professional in a scientific paper
2- line 89, the literature is not a method!
Author Response

(The authors gave the same response as above.)

Reviewer 4 Report
This paper aims to evaluate a HSR sleeper using numerical investigation. It is an interesting idea to proceed a research paper, however, a minor revision is needed before publication.
1- The novelty of research is not well reported, please elaborate.
2- Please elaborate the abaqus modeling details, such as mesh size study. and boundary condition.
3- Please use the following paper to elaborate composite sleepers properties in introduction.
"Experimental and finite element assessments of the fastening system of fiber-reinforced foamed urethane (FFU) composite sleepers"
Author Response

(The authors gave the same response as above.)

Round 2
Reviewer 1 Report
The authors have suitably revised the paper. It can be accepted as it is.
Author Response

(The authors gave the same response as above.)
